# Single-cell structural biology with intracellular electron crystallography

Štěpánka Bílá [1,2,9], Dominik Pinkas [3,9], Krishna Khakurel[4,9], Juliane Boger [5], Tomáš Bílý [1,2], Janos Hajdu[4,6], Zdeněk Franta [1], Iñaki de Diego Martinez [7], Roman Tuma [1], Lars Redecke[5,8] ✉ & Vitaly Polovinkin [4] ✉

Intracellular crystallization is an emerging approach in structural biology that bypasses the need for protein purification. In 2024, the *InCellCryst* pipeline was introduced for structural studies of intracellular crystals by serial X-ray crystallography. Serial crystallography requires the exposure of tens of thousands of cells containing intracellular crystals, precluding high-resolution studies on proteins that crystallize only in a few cells. Here we introduce *IncelluloED*, a method that combines intracellular crystallization with in situ 3D electron diffraction in cells and achieves high-resolution structures from just one crystal inside one cell. Experiments on a microcrystal of the HEX-1 protein from *Magnaporthe grisea*, grown inside an insect cell, give a structure at 1.9 Å resolution from a volume of ~1.6 μm³ as compared to 1.8 Å resolution achieved by serial X-ray crystallography from a combined volume exceeding eleven million μm³. *IncelluloED* uses widely available cryo-EM tools and brings high-resolution structural biology into home laboratories while also advancing a vision for a "single-cell structural laboratory".

Protein crystallization inside living cells occurs naturally and serves roles such as storage, protection, solid-state catalysis, and, in some cases, arises from progression of disease or infection[1]. Recombinant gene expression can also induce intracellular crystallization, and targeted *in cellulo* crystallization of proteins[1,2] is an emerging approach in structural biology. In certain cases, the quasi-native cellular environment enables binding and identification of physiologically relevant ligands and cofactors[2,3], which AI-based structure prediction cannot yet replicate. Moreover, fully glycosylated proteins, such as *T. brucei* cathepsin B, have been crystallized in living insect cells[4,5] while conventional crystallization trials were not successful. Since the first structure determination of a recombinantly produced and intracellularly crystallized protein in 2007[6], structure elucidation of *in cellulo* protein crystals has evolved significantly. The development of micro-

and nano-focus beamlines at advanced synchrotrons and X-ray free-electron lasers (XFELs), coupled with serial data collection strategies, made atomic-resolution[7] studies of tiny intracellular crystals possible. As a result, over 30 *in cellulo* structures are now deposited in the Protein Data Bank (PDB)[1,3,5,8,9].

Systematic strategies for testing crystallization of recombinant proteins in living insect cells have been developed[10,11], including the *InCellCryst* pipeline[2] where a recombinant baculovirus (rBV) is employed to drive target gene expression to increase local protein concentrations, triggering crystal nucleation and efficient crystal growth[2,4,5,12,13]. Structures are then determined via serial X-ray crystallography directly from the cells[2], avoiding purification and preserving quasi-native states. The efficiency of intracellular crystallization can be very low, and frequently less than 0.1% of the cultured cells may

[1]Faculty of Science, University of South Bohemia in České Budějovice, České Budějovice, Czech Republic. [2]Laboratory of Electron Microscopy, Institute of Parasitology, Biology Centre CAS, České Budějovice, Czech Republic. [3]Electron Microscopy Core Facility, Institute of Molecular Genetics ASCR, Prague 4, Czech Republic. [4]ELI Beamlines Facility, The Extreme Light Infrastructure ERIC, Dolní Břežany, Czech Republic. [5]Institute of Biochemistry, University of Lübeck, Lübeck, Germany. [6]Department of Cell and Molecular Biology, Uppsala University, Uppsala, Sweden. [7]European XFEL GmbH, Schenefeld, Germany. [8]Photon Science, Deutsches Elektronen Synchrotron (DESY), Hamburg, Germany. [9]These authors contributed equally: Štěpánka Bílá, Dominik Pinkas, Krishna Khakurel. ✉e-mail: lars.redecke@uni-luebeck.de; vitaly.polovinkin@eli-beams.eu

contain a protein crystal. This is a serious problem, because serial crystallography with small intracellular crystals requires the exposure of tens of thousands of individual crystals for a full data set, and the low number of cells containing a crystal may preclude structure elucidation even at advanced synchrotron and XFEL facilities. This constitutes a major bottleneck in the *InCellCryst* pipeline.

Exploiting the strong interactions of electrons with matter[14], three-dimensional electron diffraction (3D ED or MicroED)[15,16] allows structure determination from sub-micrometer-sized protein crystals, like a 2.1 Å lysozyme structure from a 0.14 μm³ crystal[17]. Developments with 3D ED have produced a number of protein and peptide structures (e.g., see refs. [15,18–22]), offering the ability to record atomic resolution diffraction data (≤1.2 Å resolution[7]) from protein crystals of sub-micrometer dimensions[23,24]. The small crystal volume and the low number of crystals required makes 3D ED a promising technique for *in cellulo* structure elucidation of proteins crystals.

The possibility of collecting ED data sets from multiple intracellular crystals in frozen cells was recently demonstrated for the major basic protein-1 (MBP-1)[25], which naturally forms numerous nanocrystals in the cytoplasmic secretory granules of eosinophils. The study also showed that combining cryo-electron tomography[26] with 3D ED enables analysis of physiologically relevant structural changes associated with the release of MBP-1 in situ and on multiple length scales, from molecular structure to nanometer-scale changes in the crystal packing during MBP-1 release.

In a typical 3D ED experiment[15,27], a protein crystal is deposited on a transmission electron microscopy (TEM) grid, vitrified, and an ED data set is recorded under cryogenic conditions as the crystal is continuously rotated under parallel illumination in a high-energy electron beam (typically 200–300 kV). Optimal sample thickness is 200–300 nm[28,29]. Thicker samples, like a cell containing a crystal, require thinning to reduce multiple scattering and absorption[15]. Thinning a crystal to a lamella of desired thickness can be performed by focused ion-beam milling under cryogenic conditions (cryo-FIB milling), using a dual-beam focused ion-beam scanning electron microscope (FIB/SEM)[26,30–33]. Protein crystals have been shown to retain crystalline order after thinning[23,31,32], making cryo-FIB a standard and reliable method for 3D ED sample preparation. However, cryo-FIB/SEM provides only topographical information and cannot localize deeply embedded crystals. Cryo-light microscopy (cryo-LM) and correlative light-electron microscopy (CLEM), established for cryo-electron tomography[26,34], offer a solution.

Here we present the *IncelluloED* pipeline, which combines the *InCellCryst* intracellular-crystallization pipeline with in situ 3D ED, enabling high-resolution structure determination from a single crystal within a single cell. A schematic overview of the *IncelluloED* workflow is given in Fig. 1. Using microcrystals of the HEX-1 protein from the filamentous fungus *Magnaporthe grisea* (MgHEX-1) grown inside High Five insect cells, we demonstrate that our approach delivers lamellae from appropriate cellular sections containing the target crystals. The lamellae are suitable for high-resolution structure determination by 3D ED and yield the previously unknown MgHEX-1 structure at 1.9 Å and 2.2 Å resolution from volumes of ~1.6 μm³ and ~0.8 μm³, respectively. For comparison, we also determine the structure of MgHEX-1 by serial synchrotron X-ray crystallography (SSX), using 62,496 crystals, and obtain a structure at 1.8 Å resolution from a total volume of about 11 million μm³. *IncelluloED* achieves similar structural details from a single crystal, overcoming the key limitation of *InCellCryst*, and opening possibilities for cellular structural sciences.

## Results

### Intracellular crystallization of MgHEX-1
To establish the *IncelluloED* approach, MgHEX-1, a structurally uncharacterized HEX-1 protein derived from the filamentous fungus *M. grisea*, has been selected. HEX-1 proteins are highly conserved among ascomycetes and self-assemble into a hexagonal crystalline core within peroxisome-derived organelles (Woronin bodies) and help to seal septal pores during cellular stress[35–37]. A *Neurospora crassa* HEX-1 (NcHEX-1) was found to produce regular, hexagonal crystals with average dimensions of 9.1 ± 3.2 μm (± standard deviation) in length and 3.5 ± 0.7 μm in width when overexpressed in insect cells using the rBV system[8,38]. This protein was quickly established as a test protein for intracellular crystal detection by various techniques (e.g., small angle X-ray scattering - X-ray powder diffraction, SAXS-XRPD[38], fixed-target serial diffraction data collection at XFELs[8] and synchrotron sources[39]) and led to the development of the *InCellCryst* pipeline[2]. The crystallization efficiency and low-micrometre size range position HEX-1 crystals as ideal candidates for *IncelluloED* method development. Moreover, the elucidation of a second HEX-1 structure can provide insights into biological self-assembly processes and into the structural conservation within this protein family. Thus, we selected MgHEX-1 as a pilot system in this study that shares 74% sequence identity with NcHEX-1.

Applying the *InCellCryst* approach, up to three hexagonal-shaped structures per cell were detected by light microscopy in approximately 40% of the rBV-infected High Five insect cells at day four post infection, exhibiting a hexagonal morphology similar to that of previously reported NcHEX-1 crystals[2,37] (Supplementary Fig. 1). The mean size of MgHEX-1 crystals, which ranged from 4 to 15 μm in the longest dimension, was estimated at 8.6 ± 2.5 μm (± standard deviation) (Fig. 2a, b). Infected High Five cells also produce enhanced yellow fluorescent protein (EYFP) as a fluorescence marker encoded on the bacmid from the DH10EmBacY cells[40], which enables 3D visualization of intracellular MgHEX-1 crystals in a negative imaging manner using fluorescence (FL) microscopy (Fig. 2c–e). This was used for 3D targeting of crystals in subsequent cryo-FIB experiments. The High Five cells containing the MgHEX-1 crystals were used both for 3D ED experiments and for X-ray serial crystallography measurements.

### Workflow of *in cellulo* crystal preparation for ED experiments
Building upon the development of cryo-FIB milling procedures for crystals[31,33,41] and also on the idea of using 3D negative fluorescence imaging for accurate 3D localization of target crystals inside a cryo-FIB/SEM microscope, we established the protocol for ED sample preparation. The procedure is shown schematically in Fig. 1 with key steps described below. Technical details are provided in the corresponding sections of "Methods" and in Supplementary Figs. 2–7.

In the first step (Fig. 1, panel 1), presence of MgHEX-1 crystals in cells was verified by light microscopy at room temperature (Fig. 2). Then, cell aliquots were collected, deposited onto a TEM grid and plunge-frozen. The frozen sample grids were sputtered with platinum prior to cryo-LM imaging (Supplementary Figs. 2 and 3). The ~20 nm metallic Pt sputter coating layer improved both reflected light contrast and subsequent FIB/SEM imaging, and assisted coordinate transfer between cryo-light and FIB-SEM platforms.

In the second step (Fig. 1, panel 2), regions of interest (ROIs) containing crystals were first identified using combined negative fluorescence (FL), reflected light (RL) and transmitted light imaging (Supplementary Fig. 2) of the frozen grids in a cryo-LM microscope. This step is of particular importance if only few cells contain crystals since the cryo-LM platform enables rapid screening of wide areas. Then, the negative FL imaging yielded 3D coordinates of crystal centers (Supplementary Fig. 3). These coordinates were further referenced to the top sample surface, established by 3D RL imaging. The RL data provided surface topography information analogous to subsequent FIB/SEM imaging and aided transfer of the crystal coordinates (Supplementary Fig. 4).

In the third step (Fig. 1, panel 3), the samples were moved into a Ga⁺ cryo-FIB/SEM microscope, and the crystal coordinates were transferred to the microscope frame by correlating the FIB/SEM

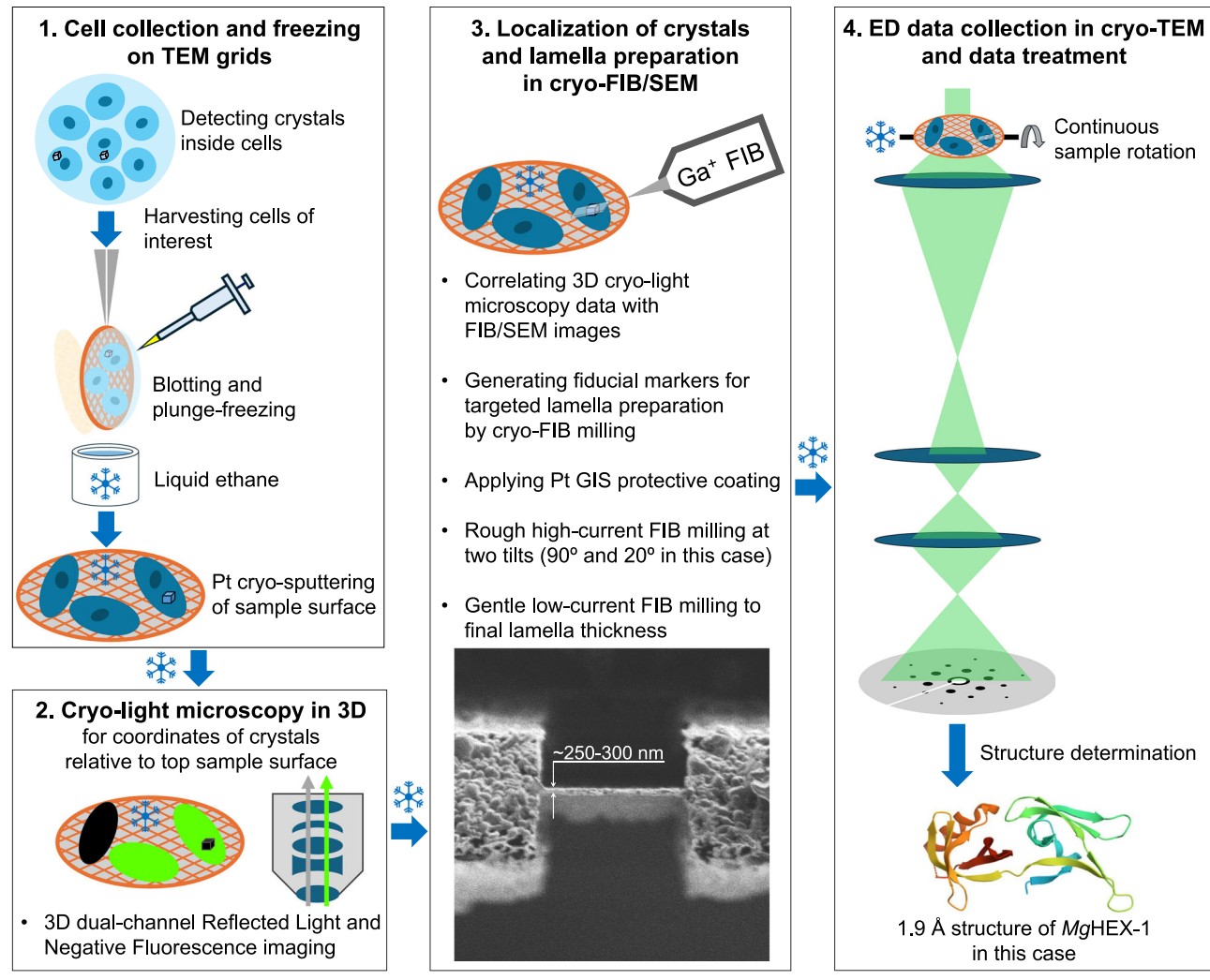

**Fig. 1 | Schematic overview of the *IncelluloED* experimental workflow.** First, intracellular protein crystals are produced and detected inside cells using the *InCellCryst* pipeline[2] (panel 1). Selected cells are collected and plunge-frozen on transmission electron microscope (TEM) grids. The top surface of the frozen samples is coated with a thin platinum layer deposited by sputtering at cryogenic temperatures. The Pt layer enhances surface contrast during subsequent reflected light microscopy, reduces charging, and protects the sample during subsequent imaging in a dual-beam focused ion-beam scanning electron microscope (FIB/SEM). The workflow continues with cryo-light microscopy (panel 2), where regions of interest and the coordinates of crystals relative to the sample surface are determined using combined 3D dual-channel reflected light and negative

fluorescence microscopy. The samples are then cryo-transferred into a cryo-FIB/SEM (panel 3), where the top sample surface is imaged. Correlation of the FIB/SEM images with the light microscopy data from panel 2 enables localization of target crystals beneath the surface. Fiducial markers are then generated on the sample using the FIB for subsequent targeted milling. At this point, an additional protective organoplatinum coating is applied using a gas injection system (Pt GIS), and crystal lamellae with a thickness of 250–300 nm are produced by cryo-FIB milling. Finally, the frozen lamellae are transferred into a cryo-TEM (panel 4), where a complete 3D electron diffraction (ED) dataset is collected from a single crystal lamella to determine the protein structure.

imaging data with the previously acquired cryo-LM data. Specifically, the ROIs containing the crystals were located by aligning the SEM overview images with the FL/RL data (Supplementary Fig. 4a). For each ROI, the RL data coupled with a 3D crystal position (relative to the top sample surface) were correlated with the FIB images of the ROI via shared surface topographical features (Supplementary Fig. 4b). This enabled accurate localization of target crystals. Fiducial markers were generated by the Ga⁺ FIB (Supplementary Fig. 5) to guide the subsequent crystal lamellae production using cryo-FIB milling.

Finally (Fig. 1, panel 3), lamellae containing parts of target crystals were prepared using a two-step cryo-FIB milling process with 30 kV Ga⁺ ions. First, ~300 nm thick protective organoplatinum coating was applied via a gas injection system (Pt GIS)[28,31,41], and rough high-current milling at 90° and 20° angles was run to yield ~4.5 μm thick lamellae. This step was designed to remove enough surrounding material to provide a wide tilt range for subsequent continuous rotation ED measurements in

a cryo-TEM (Supplementary Fig. 6). Then, a gentle low-current FIB milling at 20° (Supplementary Fig. 7) gradually decreased the lamella thickness to final values (~250–300 nm) desired for 3D ED experiments (Fig. 1, panel 4). The MgHEX-1 crystals selected for the measurements were typically in the range of 5–8 μm (see e.g., Fig. 3), slightly below the average observed crystal size of ~8.6 μm, and only the crystals fully contained within the cell were used for processing.

## Electron diffraction data collection

A lamella of an *in cellulo* grown MgHEX-1 crystal (~6 μm in size; Fig. 3) was used to assess whether the sample preparation method yields ED data for high-resolution structure determination. The lamella was prepared with a thickness of ~250 nm (Fig. 3b, c), an optimal value reported for 200 kV ED[28,29]. Following localization of the crystal in imaging mode in a 200 kV cryo-TEM (Fig. 3d), ED data were recorded on a scintillator-based electron detector (see "Methods") from

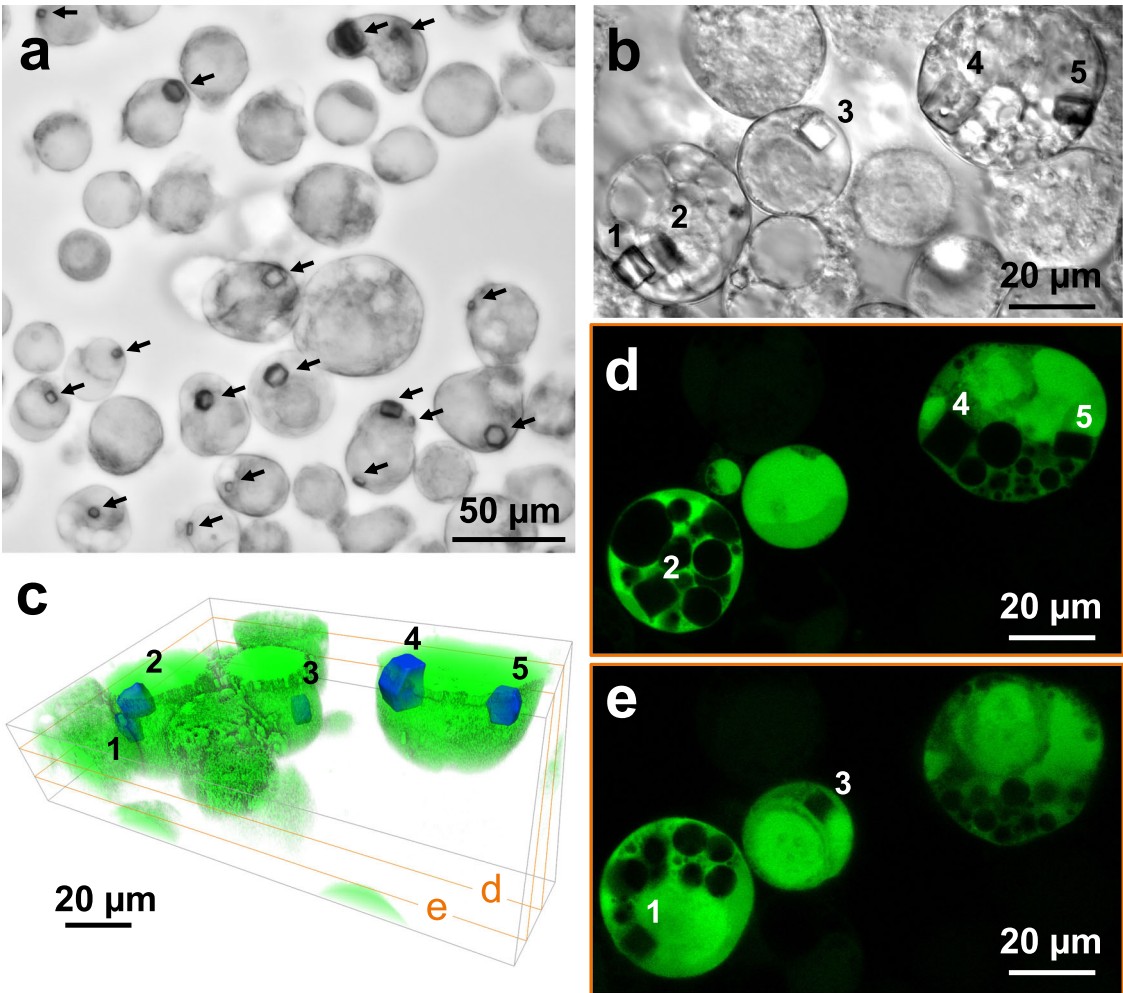

**Fig. 2 | Intracellular crystallization of MgHEX-1 in High Five insect cells and negative fluorescence (FL) imaging of the *in cellulo* crystals. a** A median intensity Z-projection of bright-field imaging Z-stacks shows the cell suspension used for ED and serial X-ray crystallography experiments. Hexagonal-shaped MgHEX-1 crystals are indicated by black arrows. **b** A median intensity Z-projection of a bright-field Z-stack of a smaller region of the cell suspension, with numbers indicating individual crystals. **c** 3D model visualization of the region in (**b**) based on a confocal FL microscopy imaging Z-stack acquired using 488 nm laser excitation at room temperature. The FL excitation of enhanced yellow fluorescent protein (EYFP), co-expressed with MgHEX-1, allows for the detection of the *in cellulo* crystals as "sharp-edged" areas lacking the FL signal. The crystals are blue-colored for enhanced visualization. The FL virtual Z-slices in (**d**) and (**e**) show areas corresponding to the MgHEX-1 crystals labelled by numbers in (**b**) and lacking the EYFP FL signal. Representative images of three independent experiments are shown.

a ~4.8 μm diameter zone with a crystal volume of ~4.4 μm$^3$ (blue circle in Fig. 3d, e). This was followed by data collection from a 1 μm diameter zone (red circle in Fig. 3d, f), corresponding to a crystal volume of only ~0.2 μm$^3$. Both 200 kV ED signals were acquired as still images with the same electron fluence of ~0.087 e$^-$/Å$^2$ (corresponding to an absorbed radiation dose of ~0.38 MGy[42]).

Further illumination of the crystal lamella with the ~4.8 μm diameter beam of 200 kV electrons was used to follow the radiation-induced decay[42–45] of the MgHEX-1 diffraction pattern (Supplementary Fig. 8). The summed intensity of high-resolution ED peaks (1.9–2.5 Å range; Supplementary Fig. 8d) exponentially dropped to 50% of its initial value at a fluence of ~1.09 e$^-$/Å$^2$ (an absorbed dose of ~4.8 MGy[42]). 50% intensity loss often sets the dose limit[44–46] in high-resolution crystallography investigations. All further measurements were far from this limit.

The sample preparation method yields crystal lamellae diffracting to high resolution even at low crystal volumes while using scintillator-based (indirect) electron detection. However, direct electron detection (DED) combined with energy filtration provides superior sensitivity and signal-to-noise ratios, yielding significantly improved ED data at lower electron fluences[24,47]. Therefore, data for the MgHEX-

1 structure determination were collected using a 300 kV microscope equipped with a DED K3 electron-counting camera in a continuous rotation mode while filtering out inelastically scattered electrons with energy loss greater than 10 eV (see "Methods" for full detail).

We produced four lamellae containing crystals ranging from 5 to 8 μm in size for the 3D ED structure determination experiments. The thickness of these four cryo-FIB milled lamellae was ~300 nm, optimal for 300 kV ED[28,29]. For two of the lamellae, 3D ED data were acquired using a 2.5 μm diameter collection zone (volume of ~1.6 μm$^3$; denoted as "microvolume") with a total fluence of ~0.3 e$^-$/Å$^2$. This corresponds to ~1.1 MGy absorbed dose, which is much below the 70% ($D_{0.7}$ ~2.5 MGy) or 50% ($D_{0.5}$ ~~4.8 MGy) radiation dose limits marked in Supplementary Fig. 8d. These two microvolume ED data sets gave very similar data processing statistics (Supplementary Table 1). One of these data sets was selected for MgHEX-1 structure determination via molecular replacement. The previously reported NcHEX-1 structure (PDB 1KHI)[37], which shares 74% sequence identity, was used as a search model. This yielded a 1.9 Å resolution crystal structure of MgHEX-1 with P6$_5$22 symmetry (Table 1 and Fig. 4a), same as previously reported for NcHEX-1[2,8,37]. The structure model was refined to 0.205/0.229 of $R_{work}$/$R_{free}$.

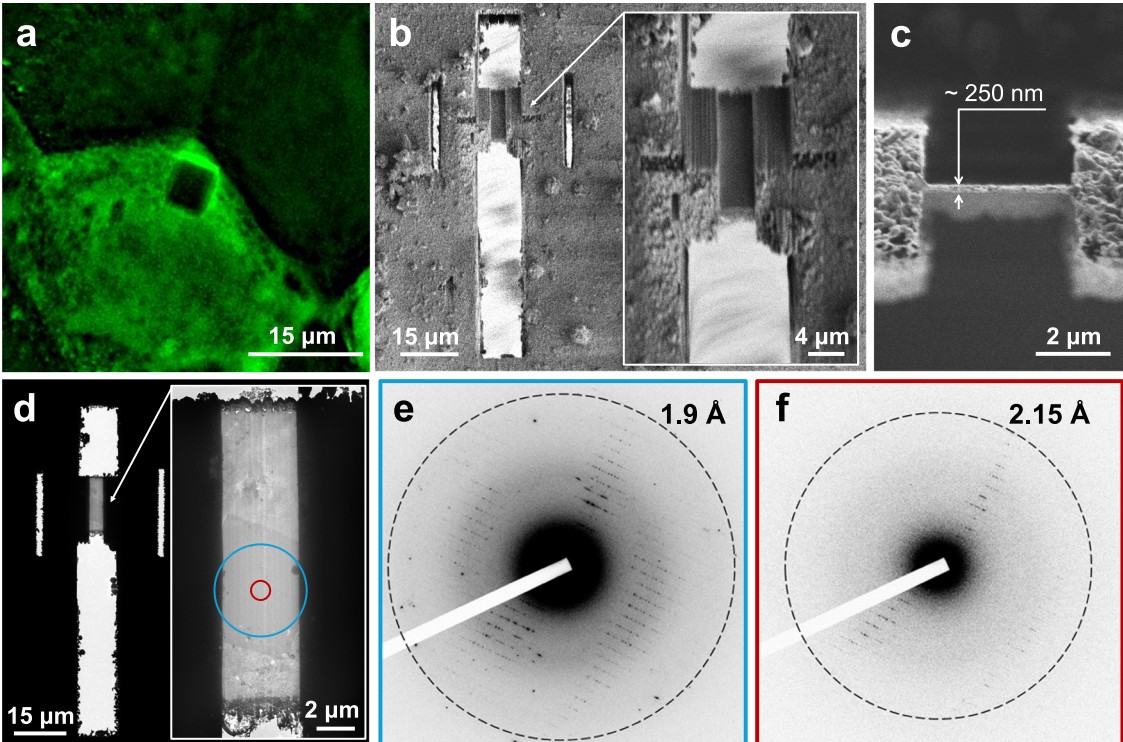

**Fig. 3 | Example of a prepared lamella containing an *in celullo* MgHEX-1 crystal and its electron diffraction (ED) characterization in a 200 kV cryo-TEM. a** EYFP-derived negative FL image of the targeted MgHEX-1 crystal, acquired around its Z-center under cryogenic conditions before the Ga⁺ FIB-milling procedure. Across more than three independent cell preparations, dozens of crystals were reliably identified on each frozen TEM grid. For ED analysis, the lamella was machined around the crystal center, had a thickness of ~250 nm, a width of ~4 µm, and was angled at 20° to the TEM grid plane. **b** 0.5 kV SEM image of the lamella prepared by cryo-FIB-milling; the inset shows an SEM image of the lamella with higher resolution. **c** 30 kV Ga⁺ FIB image of the lamella. (**d**) TEM image of the FIB-milled lamella containing a portion of the targeted MgHEX-1 crystal; the inset shows a TEM image of the lamella with higher resolution. **e** A 200 kV ED pattern obtained from a ~4.8 µm area of the lamella, defined by the parallel beam illumination and depicted by a blue circle in (**d**). The crystal volume impacting the ED signal (**e**) was estimated to be ~4.4 µm³ when excluding non-electron-transparent (black) parts in (**d**). **f** A 200 kV ED pattern collected from a 1 µm area of the crystal (crystal volume ~0.2 µm³), defined by a selected area aperture of the cryo-TEM and depicted by a red circle in (**d**). Both the ED signals were acquired at the same electron beam fluence of ~0.087 e⁻/Å².

The other two lamellae were tested using a collection zone of 1.75 µm in diameter (~0.8 µm³ volume, denoted as "nanovolume"). Three-dimensional ED data were collected from one lamella at a total fluence of ~0.6 e⁻/Å², which was double of the fluence used for the microvolume ED data collection to compensate for the reduced sample volume. This corresponds to an absorbed dose of ~2.2 MGy[42], which is still much smaller than the value of the half-intensity decay (~4.8 MGy) shown in Supplementary Fig. 8d. Processing of the nanovolume ED data resulted in a 2.2 Å resolution structure of MgHEX-1 that was refined to 0.211/0.258 of $R_{work}/R_{free}$ (Table 1 and Fig. 4b). Interestingly, the nanovolume ED signal from the second lamella reached 1.7 Å resolution. This was the highest resolution detected for all lamellae tested in this study at the same electron fluence, regardless of the crystal volume (Supplementary Fig. 9). Unfortunately, this lamella was contaminated with ice microcrystals (Supplementary Fig. 9c) during cryo-transfer into the cryo-TEM from storage, preventing reasonable 3D data collection for structure determination. Since the four crystal lamellae showed different diffraction strengths, testing more crystals could lead to improved resolution and provide better data quality in 3D ED structural studies.

During the development and optimization steps of the *IncelluloED* sample preparation workflow, we used 14 cells containing MgHEX-1 microcrystals, selected by RL/FL imaging. All crystals were successfully milled into lamellae, and all lamellae provided high-resolution ED signals. Although the localization and targeting process looks laborious, the procedures outlined in Fig. 1 yielded consistent results without loss of crystals. The weakest point in our hands was the transfer of the milled crystals into cryo-TEMs during the summer months when humidity was high in the laboratory and ice microcrystals contaminated some of the lamellae during transfer (e.g., Supplementary Fig. 9c). While this problem cannot be completely avoided, there are generally accepted strategies to mitigate it and maintain high success rates, e.g., by using fresh liquid nitrogen, filtering it through tissue or paper, and working in a low humidity environment.

## 3D ED structure of MgHEX-1

The electrostatic potential maps were refined for the 3D ED micro- and nano-volume data sets at 1.9 Å and 2.2 Å resolution (Fig. 4a, b and Table 1) and provided a high level of detail, sufficient to rebuild the non-flexible part of the main MgHEX-1 chain and to model most of the side chains. No interpretable electrostatic potential density is observed for the N-terminal residues 1–35, which have already been classified as disordered for the NcHEX-1 structure[37], and the three C-terminal residues 179-181. The maps of Gly153, Arg154 and Gly155 that are located in a flexible loop at the C-terminus of helix 1 are not fully resolved, as well as the side chains of residues Gln117 and Arg148. The final micro- and nano-volume models show a protein backbone Cα root mean square deviation (RMSD) of 0.341 Å and 0.340 Å, respectively, compared to the 134 equivalent Cα atoms of the molecular replacement search model, which is in the expected range for proteins of this sequence identity level. Moreover, the micro- and nano-volume ED models are almost superimposable with an overall RMSD of 0.166 Å for the 123 Cα atoms. Deviations of more than 0.5 Å are restricted to

**Table 1 | Data collection, processing and refinement statistics for the three MgHEX-1 crystal structures obtained in the present study**

| | Serial synchrotron X-ray diffraction (SSX) | 3D ED microvolume | 3D ED nanovolume |
|---|---|---|---|
| Number of crystals | 62,496 | 1 | 1 |
| Rotation per crystal | 1° | 80.9° | 80.8° |
| Dose per crystal (MGy) | ~1.4 | ~1.1 | ~2.2 |
| Total crystal volume (µm³) | ~11,300,000 | ~1.6 | ~0.8 |
| Sample temperature (K) | 100 | 80 | 80 |
| Space group | P6₅22 | P6₅22 | P6₅22 |
| Unit cell | | | |
| $a, b, c$ (Å) | 57.66 57.66 187.89 | 57.83 57.83 198.18 | 57.38 57.38 198.12 |
| $\alpha, \beta, \gamma$ (°) | 90 90 120 | 90 90 120 | 90 90 120 |
| Resolution range (Å) | 93.95–1.80 (1.82–1.80)* | 11.94–1.90 (1.95–1.90)* | 11.68–2.20 (2.27–2.20)* |
| Total reflections | 31,949,243 | 123,226 | 43,560 |
| Unique reflections | 18,144 (855) | 16,281 (1,053) | 9,305 (835) |
| Multiplicity | 176.9 (57.2) | 7.6 (8.0) | 4.7 (5.0) |
| Completeness (%) | 100.00 (99.88) | 99.4 (98.5) | 91.7 (92.8) |
| $I / \sigma I$ | 25.86 (0.59) | 4.8 (0.4) | 3.7 (0.5) |
| $R_{split}$ (%) | 3.34 (177.35) | – | – |
| $R_{pim}$ (%) | – | 11.3 (151.6) | 13.1 (103.3) |
| $CC_{1/2}$ | 0.9993 (0.2583) | 0.989 (0.475) | 0.973 (0.655) |
| $CC^*$ | 0.9998 (0.6407) | – | – |
| No. of collected images | 110,030 | – | – |
| No. of hits / indexed lattices | 59,042/62,496 | – | – |
| Reflections used in refinement | 17,256 (1322) | 15,389 (1076) | 8745 (603) |
| Reflections used for $R_{free}$ | 1401 (108) | 809 (55) | 499 (38) |
| $R_{work}$ | 0.2130 (0.4174) | 0.2052 (0.3170) | 0.2111 (0.3660) |
| $R_{free}$ | 0.2364 (0.4764) | 0.2292 (0.3650) | 0.2576 (0.4300) |
| No. of non-hydrogen atoms | 1126 | 1190 | 1171 |
| Macromolecules | 1033 | 1124 | 1101 |
| Solvent | 93 | 66 | 70 |
| RMS bonds (Å) | 0.009 | 0.008 | 0.005 |
| RMS angles (°) | 1.05 | 1.57 | 1.30 |
| Ramachandran favored (%) | 98.52 | 97.90 | 95.10 |
| Ramachandran allowed (%) | 1.48 | 2.10 | 4.90 |
| Ramachandran outliers (%) | 0 | 0 | 0 |
| Rotamer outliers (%) | 1.77 | 0.80 | 0 |
| Clash score | 6.81 | 4.43 | 3.18 |
| Average $B$-factor (Å²) | 40.54 | 32.10 | 51.00 |
| Macromolecules | 40.05 | 31.68 | 52.06 |
| Solvent | 45.92 | 33.64 | 48.62 |
| Mean RMSD to 1KHI (Å) | 0.520 (134 atoms) | 0.341 (134 atoms) | 0.340 (134 atoms) |
| PDB ID | 8QLX | 9RVB | 9RVD |

*Values in parentheses are for the highest-resolution shell.

residues Glu151 and Ser152 located in the flexible loop mentioned above. Even for the side chains the RMSD does not exceed 0.75 Å, except for the three residues Asn133, Glu144, Arg148 that are not fully resolved in both maps.

**X-ray crystal structure of MgHEX-1 and comparison to 3D ED structures**

Next, we compared the structural information obtained by 3D ED from a single micro- or nano-volume MgHEX-1 crystal to that obtained by applying the well-established approach of serial synchrotron X-ray diffraction (SSX) on the crystal-containing insect cells[2]. Visual inspection of test X-ray patterns revealed diffraction up to 2 Å, allowing the collection of a complete, serial line scan data set. Data processing using CrystFEL[48] and molecular replacement (MR) using again the NcHEX-1 structure (PDB 1KHI) as a search model yielded an electron density map at 1.8 Å resolution, which allowed refinement of the structure model to 0.213/0.236 of $R_{work}/R_{free}$ (Table 1 and Fig. 4c). The final MgHEX-1 structure included residues 35 to 174, again without the flexible N-terminus. In contrast to the 3D ED maps, no interpretable electron density is observed for residues 153 and 154 located in the previously mentioned flexible loop, and for the C-terminal residues 175 to 178. This might result from averaging thousands of crystals. Likewise, the side chains of residues Arg112 and Gln117 are not fully resolved. The calculated RMSD of 0.520 Å compared to the 134 equivalent Cα atoms of the MR search model is increased compared to that of 0.341 and 0.340 Å calculated for the micro- and nano-volume 3D ED structures. However, it is still in the expected range.

Comparison of the 3D ED micro- (RMSD 0.546 Å for 134 equivalent Cα atoms) and nano-volume MgHEX-1 structures (RMSD 0.509 Å for 134 equivalent Cα atoms) obtained from single crystals with that elucidated by SSX using 62,496 intracellular crystals revealed no major differences in the overall conformation of the MgHEX-1 monomer. Deviations of more than 1 Å are restricted to residues Ser66 to Gly70 located in a flexible loop connecting β-strands 3 and 4 in the N-terminal domain (Fig. 5a). Also, the calculated overall RMSD for the side chain atom positions compared to the micro- (0.738 Å) and the nano-volume (0.756 Å) model indicated an almost identical MgHEX-1 structure obtained by 3D ED and SSX. Similar RMSD values (about 1 Å) were obtained by comparing the Cα atom positions of the 3D ED microvolume and the SSX structures with that of a predicted AlphaFold 3 model[49] for MgHEX-1 (Supplementary Fig. 10).

The observed difference in the c-axes of the unit cells between the SSX structure and the 3D ED structures (Table 1) is likely attributable to the impact of distinct sample handling conditions. For SSX, cells were overlaid with PEG200 for cryo-protection during freezing in liquid nitrogen[2], which causes dehydration. For 3D ED experiments, cells were plunge-frozen in liquid ethane, which does not require cryo-protection due to a faster freezing process. A similar effect was previously observed for serial femtosecond X-ray data collection at an XFEL at room temperature[8].

All MgHEX-1 models revealed the typical two-domain structure that was previously reported for NcHEX-1[37], consisting of mutually perpendicular antiparallel β-barrels (Fig. 5b). The N-terminal domain is formed by six antiparallel β-strands, while the C-terminal domain consists of a five-stranded β-barrel and an α-helix. This indicates a high degree of structural conservation among HEX-1 proteins from different ascomycetes, especially in the "common region" comprising the 148 C-terminal residues. However, for residues 89–94, representing the most divergent part between the NcHEX-1 and the MgHEX-1 structure, the side chains fit well into the 2Fo-Fc map of the MgHEX-1 3D ED microvolume structure, in contrast to the corresponding residues of NcHEX-1 (Fig. 5c). This rules out model bias imposed by phase determination applying the molecular replacement approach with a

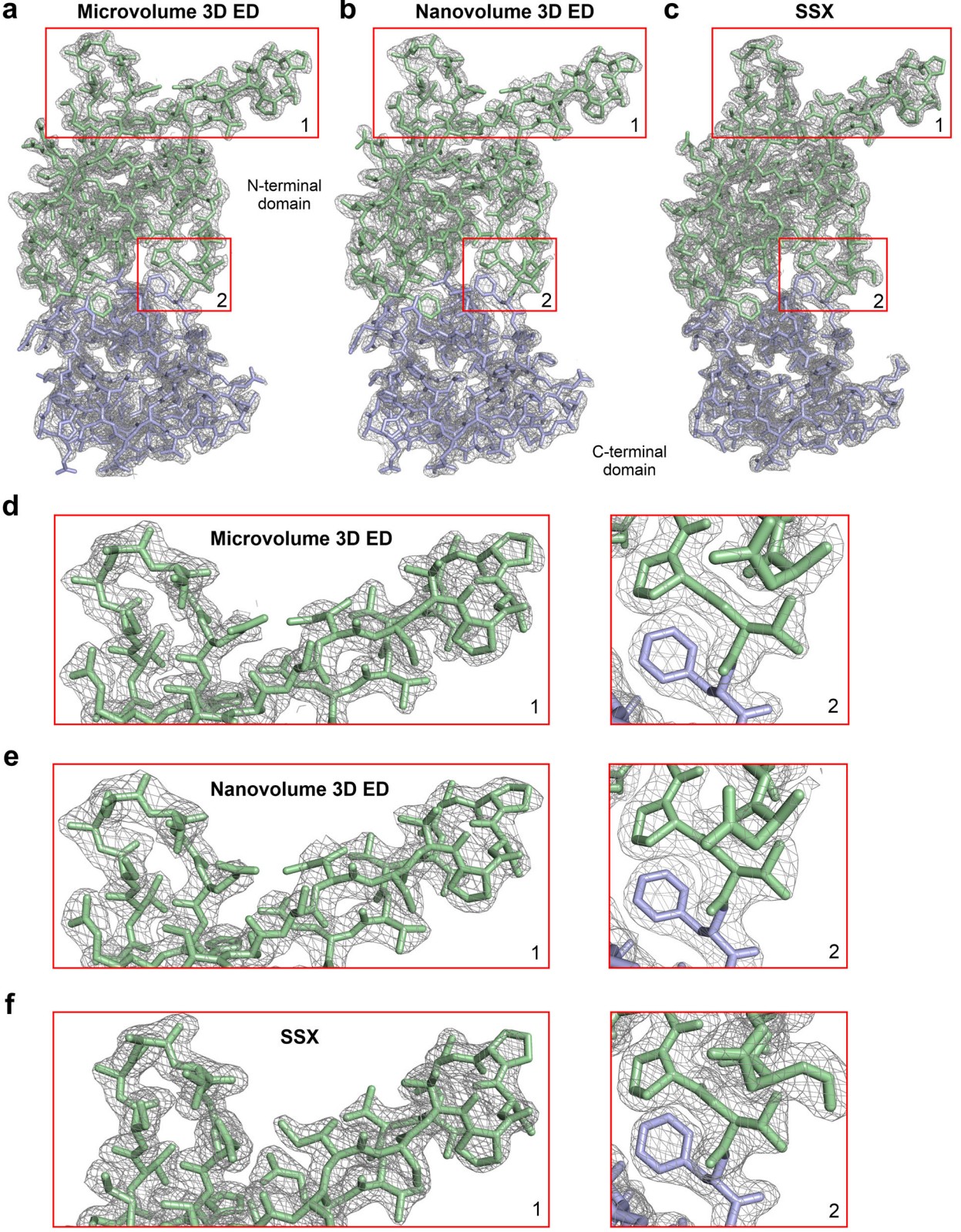

**Fig. 4 | High-quality electrostatic potential and electron density maps obtained from micro- and nano-volume 3D electron diffraction (ED) and serial synchrotron-radiation X-ray crystallography (SSX) data collected from *in cellulo* MgHEX-1 crystals.** Electrostatic scattering potential maps 2Fo-Fc contoured at 1σ (grey) calculated from the micro- (**a**) and nano- (**b**) volume 3D ED data. The modelled MgHEX-1 protein structures are shown in green (N-terminal domain) and blue (C-terminal domain) stick representation. For direct comparison, the electron density map obtained from serial X-ray diffraction of approximately 62,500 intracellular MgHEX-1 crystals is presented (**c**). Selected regions (1, residues Thr65 to His72 and Ser88 to Gln101; 2, residues Gly105 to Lys109) of the electrostatic potential and electron density maps (red squares) are shown in detail for the micro- (**d**) and the nano- (**e**) volume 3D ED data as well as for the SSX data (**f**).

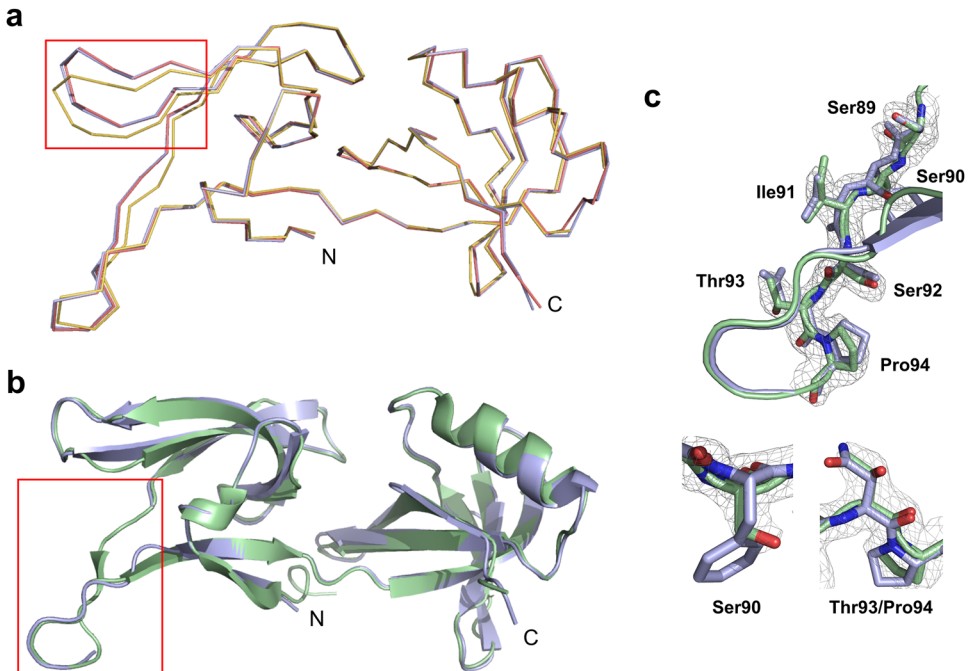

**Fig. 5 | Structural homology of HEX-1 proteins. a** Superposition of the MgHEX-1 3D ED micro- (blue) and nano- (red) volume structures with the MgHEX-1 SSX structure (yellow), all in backbone representation. The only region showing major structural differences with RMSDs above 1.0 Å is highlighted by the red box. **b** Cartoon representation of the NcHEX-1 reference structure (PDB 1KHI; green) purified from *E. coli* and crystallized by sitting-drop vapour diffusion superimposed with that of MgHEX-1 crystallized in insect cells and obtained from microvolume 3D ED (blue). The average RMSD is 0.45 Å for 123 equivalent C atoms. **c** 2Fo-Fc map (grey, contoured at 1σ) of most deviant residues 89–94 (red box in (**b**)) of the MgHEX-1 3D ED microvolume structure (green) in stick representation superimposed with the corresponding residues in the NcHEX-1 structure (blue). The side chains of residues Ser90 and Thr93 fit well into the 2Fo-Fc map calculated for MgHEX-1, in contrast to the corresponding Phe and Asn residues of NcHEX-1, ruling out phase bias imposed by phase determination applying the molecular replacement approach with a similar reference structure as the search model.

search model similar to the elucidated MgHEX-1 structure, confirming the correct interpretation of the map.

Three types of intermolecular surface interactions were previously identified that are responsible for the self-assembly of NcHEX-1 into a crystalline state in Woronin bodies[37], representing the physiological function of HEX-1 proteins in ascomycetes[36]. All involved residues are conserved in terms of sequence and structure between MgHEX-1 and NcHEX-1 (Supplementary Fig. 11a). Again, a combination of salt bridges and hydrogen bonds forms the interactions of monomers, previously denoted as group I and group II interactions, which build up a structural HEX-1 polymer that establishes a helical spiral. Each filament is in contact with six identical neighbouring polymers via hydrogen bonds (group III interactions), finally producing the six-fold symmetry of the MgHEX-1 crystals, identical to that reported for NcHEX-1[37] (Supplementary Fig. 11b).

In conclusion, the comparable level of biologically relevant information obtained by the *IncelluloED* approach presented in this study and by well-established SSX validate 3D ED as a suitable technique to obtain in situ high-resolution structural information from a single protein crystal in the very low micrometer size range. *IncelluloED* is an additional tool in the methodological toolbox, one that could be advantageous in situations where other techniques are more demanding.

## Discussion

We introduced and validated the *IncelluloED* workflow for high-resolution structure determination of intracellularly crystallized recombinant proteins, using 3D electron diffraction from a single crystal inside a single cell. As a proof of concept, we determined the previously unknown structure of the HEX-1 protein from *Magnaporthe grisea*, selected as a pilot target. The *IncelluloED* pipeline achieved comparable resolution and identical structural information to SSX while reducing the required sample volume by seven orders of magnitude to about 1 μm³, and potentially less. Its performance on other targets remains to be established, and experimental validation across a broader range of proteins will be the focus of future work.

A key question is how broadly applicable *IncelluloED* will be. We studied crystals of a fungal protein grown inside an insect cell, a heterologous host to HEX-1. Heterologous expression can present challenges, but an analysis of proteins in the PDB with identical sequences produced in native versus recombinant hosts found no cases where the same sequence adopted a significantly different global fold[50]. Post-translational modifications, host-specific chaperones, or the assembly of multi-component complexes may affect applicability and will need to be explored.

Radiation damage is a major limitation in high-resolution structural studies. Classical crystallography mitigates this by distributing damage across a large number of unit cells within a crystal. Serial crystallography with both X-rays[51,52] and electrons[53] extends this further by using tens of thousands of crystals, reducing damage proportionally. When used with intense femtosecond pulses from an XFEL, serial femtosecond X-ray crystallography can produce virtually damage-free structures by exploiting the phenomenon of "diffraction before destruction"[54]. Despite its successes, serial crystallography has limitations: averaging diffraction patterns from many crystals obscures heterogeneity; twinning can impose artificial symmetry; and conformational variation, dynamics, and rare states are smoothed out. Single-crystal methods such as *IncelluloED* avoid these issues but require frozen samples at cryogenic temperatures to reduce the effect of radiation damage on the data[44–46]. The efficiency of single crystal electron diffraction stems from the higher elastic scattering cross-sections of electrons compared to X-rays and their favourable elastic-

to-inelastic scattering ratio[14]. Considering the deposited dose, data multiplicity, and crystal volume used, 3D ED data collection in this study was over $10^5$ times more efficient than SSX (Supplementary Text 1), while using a volume 7 million times smaller. Electrons are also sensitive to charge states[15,55,56], enabling electron diffraction to map charge states of residues and cofactors[57,58].

*IncelluloED* structure determination may in some cases require data from multiple crystals. In this study, high-symmetry crystals ($P6_522$) of MgHEX-1 were used, and this enabled almost 100% data completeness (Table 1) from a single ~80° data wedge. Collecting data from low-symmetry crystals is generally more challenging[59–63]. For all space groups except triclinic P1, a 90° wedge can achieve >90% completeness, desired for reliable structure refinement and model building[60], provided the crystal is favourably oriented[59]. If necessary, one can adjust the sample orientation by rotating the grid around its Z-axis in the cryo-holder of the FIB/SEM prior to milling, based on crystal morphology and orientation on the frozen grid. These can be observed by cryo-light microscopy. P1 crystals require 180° of data, which is not possible with current electron cryo-microscopes. Practical limitations such as increased crystal lamella thickness at high tilts and the proximity of TEM grid bars (Supplementary Fig. 2) restrict usable tilt ranges to ~110°, which is less than the hardware limits of −70° to +70°[61–63]. Complete data for P1 would require at least two crystals in complementary orientations[59]. Alternatively, rotation around two complementary axes could fill a part of the missing data from a single crystal[59,64]. If high completeness cannot be achieved easily, data from multiple crystals can be merged, as is standard practice in X-ray and electron crystallography[23,45,59,65]. A common problem in ED experiments is that crystals may settle in a "preferred orientation" on the TEM grid, when for instance plate-like crystals land predominantly "face-down", creating regions of reciprocal space that are inaccessible for data collection[61–63]. This risk is low with *in cellulo* crystals, because it is the crystal-containing cell that is attached to the TEM grid support film and not the crystal itself, and orientations can be more easily randomized (e.g., Supplementary Fig. 1).

Radiation doses in this study were remarkably low (Supplementary Fig. 8d). Collecting the ~80° data wedges required only ~1.1 MGy for the microvolume and ~2.2 MGy for the nanovolume (Table 1). If data collection had continued to the 50% dose limit (Supplementary Fig. 8d: $D_{0.5}$ ~4.8 MGy for the 1.9–2.5 Å resolution shell), an additional ~270° could have been collected from the microvolume and ~95° from the nanovolume. These doses are consistent with previously reported values for global radiation damage[42,44,45] at cryogenic temperatures in protein crystals with X-rays ($D_{0.5}$ ~5–20 MGy for high-resolution data[44–46]), and those reported for electron diffraction from lysozyme[53] and proteinase K[42,43] crystals ($D_{0.5}$ ~9.4 MGy and ~5.8 MGy, respectively, for ~2 Å resolution). The doses used in our study are below those associated with site-specific radiation damage[42,44,45] from electrons, such as decarboxylation of acidic side chains (~8.8 MGy[42,43]) and breakage of disulfide bonds (~4.0 MGy[42,43]). This suggests that the *IncelluloED* approach is broadly suitable for high-resolution data collection from single crystals. For highly sensitive samples such as metalloproteins, which can be affected by doses below 1 MGy[44,45], radiation damage can be reduced by combining diffraction data from multiple crystals. This can be achieved through serial crystallography, or small-wedge data collection[51,53,65], or by dose redistribution, based on an idea described by Berglund et al.[66]. This method requires only a small number of crystals (a dozen or so) to collect full 3D data sets, each starting at systematically offset crystal orientations, but it could also be applied to a somewhat larger number of randomly oriented crystals. By selecting and combining appropriate sections from each 3D data set[66], the method redistributes the absorbed dose into full "composite data sets," from which structures at selected dose levels can be calculated. The low-dose end yields structures at kGy levels, while the high-dose end may reach several MGy. The method produces

a 3D "damage movie" and also enables off-line "resolution extension" by selecting and combining data wedges where the signal-to-noise ratios are the highest, the relative diffraction intensity changes the smallest, the unit cell dimensions remain unchanged, and data statistic parameters such as $CC_{1/2}$ and $CC^*$ are optimal[7,44–46].

From a practical perspective, the most time-consuming step was lamella fabrication by $Ga^+$ cryo-FIB milling, typically taking 2–2.5 h per crystal. New $Xe^+$ or $Ar^+$ plasma FIB systems could reduce this to under an hour[67,68]. Several workflow stages (Fig. 1), including the detection and selection of crystal-containing cells before plunge-freezing and the positioning of crystals on a grid by cryo-light microscopy, could be automated to facilitate large-scale screening of rare or very small crystals[2,69]. Automation of FIB milling could also enable unsupervised overnight operation[67]. Negative fluorescence imaging of *in cellulo* crystals requires a high numerical aperture objective to resolve small volumes and to reduce interference from the positive fluorescence of co-expressed EYFP in the surrounding media. In its current implementation (Fig. 1 and "Methods"), *IncelluloED* employs the negative fluorescence imaging and provides targeted cryo-FIB milling with a precision of ~1 μm for a crystal center. This precision, estimated by considering the ~0.7 μm Z-axis optical resolution (sub-0.4 μm in XY) and the ~300 nm Pt GIS protective coating applied during the cryo-FIB milling step, is insufficient for working with crystals of ~1 μm or smaller. The MgHEX-1 crystals used here were 5–8 μm in size, so targeted cryo-FIB milling of sub-μm crystals was not attempted and remains to be explored. Potential improvements include re-imaging lamellae after the rough milling step with enhanced 3D optical resolution, via confocal cryo-fluorescence microscopy[70,71], enabling localization precision comparable to the final lamella thickness (~250–300 nm). Alternatively, label-free imaging methods that directly probe protein crystals could improve localization. These include single- or multi-photon excited UV fluorescence[72–75], second or third harmonic generation[72,73,76], and Raman microscopy[77,78], all of which are becoming commercially available. Once implemented for cryo-EM, such methods would expand the applicability of *IncelluloED* to crystals naturally formed in cells lacking intrinsic or engineered fluorescent markers. The *IncelluloED* pipeline is built on widely available in-house instruments such as cryo-LM, cryo-FIB/SEM, and cryo-TEM. High-quality 3D ED data can already be obtained using a standard 200 kV side-entry cryo-TEM equipped with a DED camera and cryo-holders[20,55,79]. Electron energy filtering can improve the data[24,47]. Given the relatively straightforward instrument requirements, *IncelluloED* is ready to be used.

In summary, the technical developments outlined in Fig. 1 enable systematic investigations into fundamental aspects of *in cellulo* crystallization, such as the fraction of proteins that crystallize in cells, the proportion of crystals that diffract, and the identification of well-ordered mosaic blocks within a crystal lamella. Further methodological developments, such as incorporating small-molecule cofactors into host cells, may expand the potential of *IncelluloED* for high-resolution structure determination and drug discovery. Although the workflow may appear complex, the pipeline is robust, and no crystals were lost during our studies. All crystals diffracted to high resolution, although some became contaminated by ice during cryo-transfer into the microscope when the humidity was high in the lab. By eliminating the need for protein purification and requiring only a single crystal within a single cell, *IncelluloED* enables direct structure–function studies of intracellularly crystallized target proteins in a quasi-native cellular environment. This capability could be further enhanced through AI-assisted protein re-engineering to promote intracellular crystallization of complexes and functional analogues (see e.g., refs. 80–82). Using widely available cryo-EM platforms, *IncelluloED* brings high-resolution structural biology into standard laboratory settings and represents a step toward the vision of a "single-cell structural laboratory".

## Methods

*InCellCryst*, a streamlined approach for *in cellulo* crystallization of recombinant proteins in living insect cells[2], was applied to MgHEX-1, involving vector construction, virus stock production, virus stock titration, and the detection of protein crystals within insect cells.

### MgHEX-1 vector construction

Cloning of the *Hexagonal peroxisome 1 (HEX-1) gene* from ascomycetous fungus *M. grisea* (NCBI Accession AY044846, codon optimized for *Trichoplusia ni* expression) into the pFastBac1 V2 cyto vector under the control of the polyhedrin promotor[2,83] was performed by PCR amplification using primers 5′-GATCGGTACCTACTATGAAGACGATCGTGA-GACC-3′ and 5′-GATCGCTAGCCAGCCTGCTGCC GTG-3′, adding *Kpn*I and *Nhe*I restriction sites at 5′ and 3′ ends of the MgHEX-1 gene without start and stop codon and HiFi DNA polymerase (highQu). After digestion of the PCR product with FastDigest *Nhe*I and *Kpn*I (Thermo Fisher Scientific), the insert was ligated into the equally digested pFB1 V2 cyto vector by T4 DNA ligase (Thermo Fisher Scientific), yielding pFB1 MgHEX-1 V2 cyto plasmid. Competent *Escherichia coli* DH5α cells (NEB C2987) were transformed with ligation mixture and amplified pFB1 MgHEX-1 V2 cyto plasmid DNA was extracted using the GeneJET Plasmid Miniprep Kit (Thermo Fisher Scientific) and sequence verified (LGC Genomics).

### Recombinant baculovirus stock production and titration

For rBV production *E. coli* DH10EmBacY cells (Geneva Biotech) were transformed with pFB1 MgHEX-1 V2 cyto according to the Bac-to-Bac manual (Invitrogen). Recombinant bacmid DNA, isolated by applying the ZR Bac DNA Miniprep Kit (Zymo Research), was verified for the presence of the transposed pFB1 sequence by PCR using pUC/M13 primers. In a 12-well cell culture plate, $5 \times 10^5$ *Spodoptera frugiperda* Sf9 insect cells (11496015, Thermo Fisher Scientific) were grown in serum- and protein-free ESF921 cell culture medium (Expression Systems), supplemented with penicillin/streptomycin (Sigma Aldrich), at 27 °C and transfected with recombinant bacmid DNA using Escort IV reagent (Sigma Aldrich). After four days of incubation at 27 °C, the supernatant was harvested (P1 virus stock) and used for infection of $2 \times 10^6$ Sf9 cells/mL in a T-25 mL cell culture flask that was incubated for another 4 days at 27 °C with constant shaking (100 rpm), followed by harvesting the supernatant (P2 stock). The virus titre of the P2 stock was calculated using the $TCID_{50}$ (tissue-culture infectious dose[84]) in a serial dilution assay as previously described[2,38,83]. The efficiency of transfection and viral stock production as well as the virus titre was determined by detection of the fluorescence of the EYFP marker encoded in the bacmid[2] using a widefield inverted X81 (Olympus) or Ts2R-FL (Nikon) microscope.

### Intracellular crystallization of MgHEX-1

For intracellular crystallization, $0.9 \times 10^6$ High Five insect cells (B85502, Thermo Fisher Scientific) were plated in 2 mL ESF921 medium (Expression Systems) in one well of a six-well cell culture plate and infected with the MgHEX-1 encoding rBV using a multiplicity of infection (MOI) of 1. The required amount of P2 stock was calculated using the following formula: virus stock [mL] = (MOI · cell number)/(0.69 · $TCID_{50}$/mL)[85]. After 4 days of incubation at 27 °C, the High Five cells were examined for the presence of MgHEX-1 protein crystals by light microscopy via bright-field imaging[2] using Ts2R-FL (Nikon) and DMi8 (Leica Microsystems) inverted microscopes equipped with 20× and 40× objectives, respectively. Median intensity Z-projection of bright-field Z-stacks were performed by ImageJ[86] to better visualize the crystals. Cells containing MgHEX-1 crystals were carefully harvested by gentle pipetting for subsequent electron or X-ray crystallography experiments. Note that the excitation of EYFP fluorescence (FL), which is co-expressed in the cells, provided an additional way of detecting *in cellulo* crystals as regularly shaped areas lacking the fluorescent signal

(Fig. 2). At room temperature, the latter was performed with a confocal laser scanning FV1000 microscope (Olympus) using a 488 nm laser and a 100× objective (UPlanSApo, oil-immersion, NA 1.4, Olympus). 3D model visualization based on the FL stack was created using Amira software (Thermo Fisher Scientific). In Amira, the Volume Rendering function and the 2D Median filter were used, and the blue model of the crystals was semi-automatically generated (Fig. 2).

### *In cellulo* crystal preparation for ED experiments

The sample preparation workflow for 3D ED data collection is depicted in Fig. 1, and the individual steps are detailed in separate sections below.

**Cell collection and freezing on TEM grids.** After verification by light microscopy (Fig. 2) a 3 μL aliquot of cells containing MgHEX-1 crystals was carefully collected with a pipette from the bottom of a well in a six-well cell culture plate. Then, an EM GP2 plunge freezer (Leica Microsystems) was used, with its sample environment chamber set to 20 °C and 95% relative humidity. The cell aliquot was deposited onto a holey gold film covered TEM grid (UltrAuFoil® R2/2, 200 gold mesh, Quantifoil Micro Tools). The grid was then blotted with filter paper (#1, Whatman) from the back side for 12 s, directly plunge-frozen into liquid ethane cooled by liquid nitrogen bath, transferred to liquid nitrogen and clipped into an Autogrid assembly (Thermo Fisher Scientific) for further handling. The grid with the cells was stored in liquid nitrogen until further use. Note that the holey gold film covered TEM grids had better mechanical stability during the blotting procedure than holey carbon film covered grids (QUANTIFOIL®, Quantifoil Micro Tools), and this resulted in larger intact film areas with retained cells in our hands. The frozen samples were transferred, under cryogenic conditions, using a VCM/VCT500 cryo-transfer system (samples were kept below -170 °C, Leica Microsystems), into an ACE600 cryo-coater (Leica Microsystems). The samples were sputter-coated with a ~20 nm platinum conductive protection layer[41,87], while maintained below −153 °C.

**Determining regions of interest and crystal coordinates by cryo-light microscopy (cryo-LM).** The coated samples were transferred to a cryo-fluorescence upright wide-field microscope (THUNDER Imager EM Cryo CLEM, Leica Microsystems) equipped with a 50× objective possessing a high NA of 0.90 (HC PL APO, CRYO CLEM, Leica Microsystems; ~373 nm XY-resolution and ~679 nm Z-resolution at 550 nm wavelength) and a cryo-stage (temperature set to −180 °C), which accommodates up to two TEM Autogrids. Within the THUNDER microscope, the sample grid plane is oriented perpendicular to the optical axis during cryo-light microscopy (cryo-LM) imaging.

Grid overviews (Supplementary Fig. 2a) in reflected light (RL; using a LED3 fluorescence light source (Lumencor) and a longpass filter >425 nm) and FL (with 450-490 nm excitation and 500−550 nm emission filters) channels were obtained within the THUNDER cryo-LM microscope using the embedded Navigator module of LAS × software package (Leica Microsystems) by stitching individual fields of view (271.08 μm × 271.08 μm). Zones with cells exhibiting EYFP FL and an intact support film were investigated in RL, FL and transmitted light (TL; using light of a halogen lamp) regimes and regions of interest (ROIs) containing suitable crystals were identified (see Supplementary Fig. 2b).

For each ROI containing a crystal of interest, a 3D volume was imaged as a ~40 μm dual-channel Z-stack using RL and FL channels (Supplementary Fig. 3) and a system-optimized Z-step size of 0.445 μm. THUNDER deconvolution[88] technique embedded in the LAS X software was applied to the FL Z-stack data to improve the contrast of the FL images. This dual-channel RL/FL Z-stack was used to locate the 3D center of the crystal ($X_c$, $Y_c$, $Z_c$), as shown in Supplementary Fig. 3. Using the FL data of the Z-stack, the position ($X_c$, $Y_c$, $Z_c$) was

determined as the center of the negative FL signal of the crystal (Supplementary Fig. 3f). The $Z_c$-depth of the crystal's center was calculated relative to the Z-position of the top surface of the sample ($Z_s$) at ($X_c$, $Y_c$) location. The $Z_s$-position was determined using the RL Z-stack data and set as the Z = 0 reference (Supplementary Fig. 3e). The determined lateral position of the crystal center ($X_c$, $Y_c$) was marked with a dot on the maximum intensity Z-projection of the RL channel of the ROI containing the crystal (see Supplementary Fig. 3c and Supplementary Fig. 4b). This RL image and the Z-depth of the crystal's center ($Z_c$) were used for subsequent image correlation and crystal localization procedures as depicted in Supplementary Fig. 4b and Supplementary Fig. 5.

Note that the metallic Pt sputter coating of the samples, in addition to mitigating charging and protecting against unwanted ion beam damage during subsequent FIB/SEM imaging[41,87], also amplified the light reflection from the coated top surface of the samples. This enhanced the surface imaging contrast for RL microscopy and improved FIB/SEM surface imaging, thereby facilitating both the determination of the $Z_s$-position of the sample top surface (Supplementary Fig. 3c, e, g) and subsequent correlation between FIB and RL images inside the FIB/SEM microscope (Supplementary Fig. 4b). We observed that ~20 nm thick Pt sputter layers provided good light reflection while remaining sufficiently transparent for TL and FL signals, while thicker layers (e.g., ~40 nm) significantly dampened FL imaging.

**Localization of crystals in cryo-FIB/SEM and generation of fiducial markers.** Grids containing the crystals of interest were unloaded from the cryo-LM microscope and, using a VCM/VCT500 cryo-transfer system (<−170 °C, Leica Microsystems), placed into a 40° pre-tilted cryo-FIB/SEM grid holder (Leica Microsystems), which accommodates up to two TEM Autogrids per session, and inserted into an Amber cryo-FIB/SEM microscope (Tescan) equipped with a passively cooled cryo-stage (maintained at <−143 °C, Leica Microsystems). The angle between the ion and electron beams is 55° for the Amber FIB/SEM microscope. To avoid unwanted sample damage during navigation and monitoring of the milling process, SEM and FIB imaging were performed with 0.5 kV electrons of 10 pA current and with 30 kV Ga$^+$ ions of 10 pA current throughout the described cryo-FIB milling procedure.

In the cryo-FIB/SEM microscope, an overview SEM image of a TEM grid was acquired with the electron beam at a 65° angle to the grid plane (the default angle for the Autogrid-compatible sample holder used). To enable rapid localization of ROIs containing crystals of interest, this overview SEM image was correlated with the combined RL and FL maps derived from cryo-LM imaging (Supplementary Fig. 4a). This was performed using a three-point image correlation procedure (based on image affine transformations) of the CORAL module within the Essence software (Tescan), which controls the FIB/SEM instrument.

Subsequently, for each ROI containing a target crystal, the following steps were performed. An ROI with a crystal of interest was positioned at the intersection of the SEM and FIB beam foci, and the sample was rotated such that the ion beam was at a 90° angle to the sample grid plane. The ROI was then imaged by FIB, and the resulting image was correlated, using the three-point correlation procedure, with the previously obtained LM image (containing the marked lateral position of the crystal's center ($X_c$, $Y_c$); see Supplementary Fig. 4b). Using the DrawBeam module of the Essence software, fiducial markers were drawn typically as two $2 \times 5\,\mu m$ rectangles separated by a $12\,\mu m$ gap on the correlated overlay of FIB and LM images, positioning the crystal's center midway between the fiducials (Supplementary Fig. 5a, b). These fiducial markers were milled (Supplementary Fig. 5c) by raster scanning over the drawn pattern using a 30 kV ion beam with a 50 pA current to a depth of ~1 μm. For the calculations herein, we used an experimentally derived cryo-FIB milling rate of ~4 μm³/nA/s.

Subsequently, the ROI was positioned such that the ion beam was at a 20° angle to the sample plane and then imaged by FIB. The position of the crystal's center at the 20° angle FIB view was estimated to be $Z_c \times \cos 20°$ lower than the top surface above the crystal center (Z = $Z_s$) defined at this FIB view by the previously milled two fiducial markers (see Supplementary Fig. 5d–f). The third fiducial marker (a $3 \times 12\,\mu m$ rectangle) was drawn 3.25 μm lower than the estimated position of the crystal's center and FIB-milled at 150 pA current to a depth of ~2 μm. All three fiducial markers per each crystal served as 3D locators of estimated crystal center positions for subsequent procedures.

**Crystal lamella production by cryo-FIB milling.** After creating all the fiducial markers for all the crystals of interest, the samples were coated with a protective (see e.g., refs. 28,31,41) organoplatinum layer using a gas injection system (Pt GIS, 60 °C, 45 s, ~400 nm/min layer deposition rate) integrated into the Amber cryo-FIB/SEM microscope. The resulting protective Pt GIS layer was ~300 nm thick. For each crystal of interest, the following FIB milling steps were performed.

A center of a crystal of interest was relocalized at different FIB view angles using the previously FIB-marked fiducials. "Rough" cryo-FIB milling was performed via raster scanning with a 30 kV Ga$^+$ ion beam, first, at 90° and then at 20° to the TEM grid plane using high currents of 2.5 nA and 1 nA, respectively, as depicted in Supplementary Fig. 6. The bulk of the sample was removed to provide a wide rotation range accessible for transmission of an electron beam in subsequent 3D ED measurements. This high-current milling produced a ~4.5 μm thick lamella at a 20° angle to the grid plane. Subsequently, a "gentle" cryo-FIB milling procedure was performed by raster scanning with the 30 kV Ga$^+$ FIB at 20° to the TEM grid plane in a stepwise manner, reducing the beam current as the lamella was thinned to its final thickness, as shown in Supplementary Fig. 7. Milling currents of 150, 50, and 10 pA were used (in this order) to generate the final lamella by decreasing the thickness of the sample to ~1.5 μm, ~0.8 μm, and the final thickness, respectively. Such low currents were used for the gentle procedure to retain the maximum possible degree of crystalline order of the *in cellulo* crystal (see e.g. refs. 31,87). Lamellae produced for 200 and 300 kV ED experiments were ~4 μm wide and with final thicknesses, corresponding to the reported optimum[28,29], of ~250 nm and ~300 nm, respectively, and were angled at 20° with respect to the TEM grid plane.

Using the VCM/VCT500 cryo-transfer system, the grids were then transferred from the cryo-FIB/SEM microscope to cryo-EM grid boxes (Thermo Fisher Scientific) for storage in liquid nitrogen until further use in ED experiments.

**ED data collection, data processing and structure determination**
For 200 kV ED experiments, we used a JEM-F200 cryo-TEM (JEOL) equipped with a cold cathode field-emission electron gun, a side-entry dual Autogrid cryo-holder (-177 °C, Model 210, Simple Origin), and a scintillator-based CMOS detector (4096 × 4096, $15.5 \times 15.5\,\mu m^2$ pixel, TemCam-XF416, TVIPS). Crystal lamellae were identified in imaging mode (Fig. 3d), and ED experiments were performed with parallel electron beam illumination at a flux of ~0.0435 e$^-$/Å$^2$/s. ED data were collected either from the parallel beam illumination area, which was ~4.8 μm in diameter on the sample plane, or from a crystal lamella zone of ~1 μm in diameter, as defined by a selected area aperture. ED data were recorded as still images with the XF416 camera, using $2 \times 2$ binning and a 2 s exposure (corresponding to a fluence of ~0.087 e$^-$/Å$^2$). The effective sample-to-detector distance was ~2310 mm, as calibrated with a standard evaporated aluminum grid (Ted Pella). To evaluate radiation sensitivity of a MgHEX-1 crystal lamella, the sample was irradiated with the ~4.8 μm diameter 200 kV electron beam (Fig. 3d), and a series of 40 still ED images was collected sequentially. The resulting total cumulative fluence was ~3.5 e$^-$/Å$^2$. The intensities of high-resolution ED peaks (1.9–2.5 Å range) were then summed for each

image and analyzed as a function of the cumulative electron dose (Supplementary Fig. 8). ED data visualization and analysis were performed using Adxv[89], ImageJ[86] software and an in-house Python 3.12.4 script. Absorbed radiation doses were estimated with the RADDOSE-ED software[42].

For 300 kV ED experiments, we used a Titan Krios G1 cryo-TEM (Thermo Fisher Scientific), equipped with a Schottky field emission gun (XFEG), a cryogenic sample Autoloader (--193 °C), a GIF Bio-Quantum energy filter (Gatan/Ametek) and a K3 direct electron detector (5760 × 4092, 5 × 5 μm² pixel, Gatan/Ametek). The sample grids were cryo-transferred from storage to the Autoloader cassette, ensuring the microscope's rotation (α-tilt) axis was orthogonal to the FIB milling direction. In the cryo-TEM, the crystal lamellae were identified in imaging mode (Supplementary Fig. 9). ED data acquisition employed a parallel electron beam with a 14 μm diameter illumination area on the specimen plane and a low electron flux of -0.0009 e⁻/Å²/s. The energy filter slit was set to select only electrons with energy loss less than 10 eV. Following the manufacturer's recommendations (Gatan/Ametek) for 3D ED data collection, the K3 detector was operated in standard electron-counting mode (without correlated double sampling; internal frame rate of 1.5 kHz) at a dose rate below 40 e⁻/pixel/s for ED peaks (to avoid significant coincidence loss[90,91]). This dose rate was achieved by setting up the low flux of the parallel electron beam (-0.0009 e⁻/Å²/s) and validated by acquiring 1 s exposure ED still images from a crystal lamella prior to 3D ED collection. The effective sample-to-detector distance was -905 mm, as calibrated with a standard evaporated aluminum grid (Ted Pella). 3D ED data were recorded in a continuous rotation mode collecting ED signal from crystal lamella zones with diameters of 2.5 μm (referred to as "microvolume") or 1.75 μm (referred to as "nanovolume") as defined by selected area apertures of the cryo-TEM. The K3 camera recorded at 1 fps during the data collection. Microvolume data were collected during 325 s exposure at a rotation speed of 0.249°/s, covering a total angular wedge of 80.9° (from −40.4° to +40.5° α-tilt) at a fluence of -0.3 e⁻/Å². Nanovolume data were collected during 660 s exposure at a rotation speed of 0.1224°/s, covering a total angular wedge of 80.8° (from −40.4° to +40.4° α-tilt) at a fluence of -0.6 e⁻/Å². Absorbed radiation doses were estimated using RADDOSE-ED software[42]. The electron-counting data images were saved as 8-bit unsigned Tagged Image File Format stack without detector gain normalization being applied. For further processing, the images were gain-normalized and converted to the Crystallographic Binary File format using an in-house developed Python 3.12.4 script and 2cbf program of XDS software package[92].

The 3D ED data were indexed and integrated using XDS software[92], and then scaled and merged using AIMLESS software[93]. Initial phases were obtained by molecular replacement with Phaser[94] using the NcHEX-1 structure, previously obtained using crystals grown by a sitting drop vapor diffusion method, as a search model (PDB 1KHI). MgHEX-1 crystal structure refinements for all collected data sets were carried out using REFMAC5[95] (version 5.5.0051; electron scattering factors were set with the keyword "source EC MB") and iterative cycles of manual model building in Coot[96] (version 0.9.7). PyMOL Molecular Graphics System (version 4.5.0, Schrödinger, Inc.) was used for graphical illustrations.

### Serial X-ray diffraction data collection, data processing and structure determination

Sample preparation for X-ray diffraction measurements was performed as previously described[83]. In brief, High Five cells carrying MgHEX-1 protein crystals were harvested into a 1.5 mL tube and allowed to settle by gravitation. In a 90% humidity air stream 0.5 μL cells from the loose pellet were loaded onto a MicroMesh (700/25; MiTeGen) previously mounted on a B1A goniometer base (MiTeGen), which was positioned by a custom-made holder in the optical focus of a standard upright cell culture light microscope. The excess medium was removed using an extra fine liquid wick (MiTeGen). For cryo-protection, 0.35 μL of a 40% PEG200 solution diluted in ESF921 cell culture medium were pipetted onto the cells, again followed by removal of the excess liquid. Cell-loaded MicroMeshes were flash frozen and stored in liquid nitrogen until further use in X-ray diffraction experiments.

X-ray diffraction data collection was performed at the EMBL microfocus beamline P14 at the PETRAIII storage ring (DESY, Hamburg) using the mxCuBE v2 user interface[97]. The cell-loaded Micro-Meshs were mounted on a mini-Kappa goniostat attached to an MD3 diffractometer using the MARVIN Sample Changer. Using a 7 × 3 μm² double focused X-ray beam characterized by an energy of 12.7 keV and a flux of 1.71 × 10¹³ ph/s MicroMeshes were scanned employing a previously established serial helical grid scan strategy[2] to collect complete data sets using the EIGER 16 M detector (DECTRIS) at 100 K. During 20 ms irradiation at each position the MicroMesh was rotated by 1°. Absorbed radiation doses were estimated using RADDOSE-3D software[42]. Data collection parameters are presented in Table 1.

To process X-ray diffraction data using CrystFEL[48], a geometry file specifying information about the sample to detector distance (clen), the wavelength (photon_energy), the size of the detector (max_fs and max_ss) as well as the beam position on the detector relative to the detector corner (corner_x and corner_y) is required. Then, peak detection parameters can be specified using the graphical user interface of CrystFEL version 10.0/10.1 or the check-peak-detection script of version 9.1 and older versions using peak finding algorithm peakfinder8. In general, for this study, a threshold of 0, a local_bg_radius of 3, min_res of 50, a max_res of 2200 and a max_pix value of 50 were a suitable starting point. The min_snr and min-pix were adjusted to allow proper peak detection for each sample. Afterwards, all frames were indexed by Mosflm, and the correct lattice and unit cell parameters were optimized, and cycles of beam position refinement were performed by executing the detector_shift script (version 9.1) on the obtained stream files. If the optimized beam position did not differ more than 0.1 mm from the input, the beam position was accepted and all frames were indexed invoking mosflm-latt-cell, mosflm-latt-cell, xds and xgandalf. Then the peakogram-stream script was executed to set the maximum allowed intensity for each resolution range to separate salt reflections from protein reflections applying a filtering script. Finally, the filtered intensities were scaled and merged. The resolution limit was set where CC* is above 50%, SNR above 0.5 and the completeness over 95%. Mtz-files for subsequent modeling were generated using get_hkl. To set up peak detection parameters CrystFEL version 10.0 was applied, for indexing CrystFEL version 10.1 or 10.2, while for merging and scaling CrystFEL version 9.1 or 10.2 were used.

Phases were retrieved by molecular replacement with Phaser[94] using the NcHEX-1 structure as a search model (PDB 1KHI). Structure refinements for all generated data sets were carried out using PHENIX[98] (version 1.19.2-4158) and iterative cycles of manual model building in Coot[96] (version 0.9.7). Simulated-annealing omit maps were calculated in PHENIX. Applying the FastFourierTransform program, electron density maps with CCP4 extensions were saved and loaded in the PyMOL Molecular Graphics System (version 4.5.0, Schrödinger, Inc.) for graphical illustrations of contoured omit maps.

### Reporting summary

Further information on research design is available in the Nature Portfolio Reporting Summary linked to this article.

## Data availability

The atomic coordinates and structure factors of the SSX structure as well as of the microvolume and nanovolume 3D ED/MicroED structures of MgHEX-1 have been deposited in the RCSB Protein Data Bank (PDB) under accession codes 8QLX (MgHEX-1 SSX structure), 9RVB

(MgHEX-1 microvolume 3D ED/MicroED structure) and 9RVD (MgHEX-1 nanovolume 3D ED/MicroED structure), respectively.

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

## Acknowledgements

We thank Sahil Gulati (Gatan/Ametek) and Samuel Záchej (Tescan) for helpful technical discussions regarding the K3 camera operation and the FIB/SEM microscope operation, respectively. We acknowledge the Electron Microscopy Core Facility, IMG, Prague, Czech Republic, supported by the MEYS CR (LM2023050) and the European Regional Development Fund (No. CZ.02.1.01/0.0/0.0/18_046/0016045, No. CZ.02.01.01/00/23_015/0008205), for their support with experiments and data collection performed at cryo-LM, FIB/SEM and 200 kV TEM microscopes. We acknowledge Cryo-electron microscopy and tomography core facility CEITEC MU of CIISB, Instruct-CZ Centre, supported by the MEYS CR (LM2023042) and the European Regional Development Fund (Project "Innovation of Czech Infrastructure for Integrative Structural Biology"; No. CZ.02.01.01/00/23_015/0008175), for their support with 300 kV TEM experiments. We thank Jiří Nováček for his assistance with these experiments. The synchrotron diffraction data was collected at the P14 beam line operated by EMBL Hamburg at the PETRA III storage ring (DESY, Hamburg, Germany). We thank Gleb Bourenkov for the assistance in using the beamline. Š.B., T.B., Z.F. and R.T. have been supported by the European Regional Development Fund (No. CZ.02.1.01/0.0/0.0/15_003/0000441). Š.B. and R.T. was supported by the Czech Science Foundation (GACR; project No. 25-17423). J.H., K.K. and V.P. acknowledge support from the Czech Science Foundation (GACR) project No. 24-10671S, and by a grant from the Johannes Amos Comenius Programme of the Ministry of Education, Youth and Sports of the Czech Republic through the SENDISO project No. CZ.02.01.01/00/22_008/0004596. K.K. and V.P. also thank the European Union's HORIZON EUROPE framework programme for research and innovation (Grant agreement No. 101094299 "IMPRESS"). L.R. thanks the German Federal Ministry for Education and Research (BMBF; grant 05K18FLA) for the support.

## Author contributions

V.P. and L.R. conceived the *IncelluloED* approach, which was designed with Š.B., D.P., K.K., J.H., T.B., Z.F., I.D.M., and R.T. Š.B. and J.B. performed *in cellulo* crystallization experiments, supervised by R.T., Z.F., and L.R. Light microscopy, sample blotting, FIB/SEM, and TEM experiments were performed by Š.B., D.P., K.K., V.P. with the help of T.B., R.T., Z.F., and were generally overseen by V.P. K.K. and V.P. processed the ED data and performed molecular replacement with the help of J.H. L.R. and V.P. refined the ED structures, and calculated the electrostatic potential maps. Supervised by L.R., J.B. performed SSX experiments, processed the SSX data, performed molecular replacement, refined the X-ray structure, and calculated the electron density map. The manuscript was prepared by Š.B., D.P., K.K., J.H., Z.F., R.T., L.R. and V.P. with input, discussions and improvements from all authors.

## Competing interests

The authors declare no competing interests.
