## [Transparent Peer Review file · Nature Communications]

A single-crystal electron diffraction pipeline for *in situ* protein structure determination inside cells

Corresponding Author: Dr Vitaly Polovinkin

Version 0:

Reviewer comments:

Reviewer #1

(Remarks to the Author)

In this manuscript, Redecke and co-workers present an adaptation of the InCellCryst concept, (i.e. a pipeline based on crystallization directly within living insect cells to produce a sufficient amount of homogeneous microcrystals of a target protein for structure determination by serial X-ray, thereby avoiding the purification step), to electron diffraction (InCellED), a technique that is gaining momentum but is still in its infancy in the context of protein crystallography.

They chose HEX-1 from *Magnaporthe grisea* (MgHEX-1) crystals, grown in insect cells to preserve their native state, to demonstrate that electron diffraction data collected from a single crystal can yield enough information to determine the previously unknown 3D structure at 1.9 Å resolution. They also compare these results with a similar dataset obtained by serial X-ray crystallography, which used tens of thousands of crystals and reached a resolution limit of 1.8 Å. Because electron diffraction requires far fewer cell-containing crystals, this method could help overcome the bottleneck of low efficiency in in cellulo crystal growth. Nevertheless, sample production, preparation, and adaptation for 3D microED data collection require specialized know-how, which is still under development.

The chosen system, MgHEX-1, is consistent with their previously characterized *Neurospora crassa* HEX-1 (NcHEX-1), since both naturally self-assemble into a hexagonal crystalline material. The new MgHEX-1 structure could shed light on the mechanism underlying the self-assembly process that drives cellular stress but further analysis seems to be needed to find the mechanism behind its formation naturally.

The procedure involves cryogenic focused ion beam milling (cryo-FIB milling) coupled to scanning electron microscopy (SEM) to adjust sample thickness after locating crystals within the cell using fluorescence (FL) microscopy, as well as cryo-light microscopy and correlative light–electron microscopy. The sample preparation process is well described; however, fine details (beyond the scope of the present work) are not provided, and reproducibility may be challenging even for experienced technicians.

Overall, the manuscript is well written, and the methodology, its main focus, is clearly explained, even from the perspective of a non-expert in electron diffraction. Therefore, I believe this article will be significant for the structural biology community, although I have one major concern regarding its framing.

Major concerns:

Although this approach for InCellED clearly has high potential and could become a viable alternative technique, the data presented in the manuscript do not allow one to envision its general applicability. Looking at the authors' own previous work in this journal, it is evident that they have several other systems available that could be used to showcase the broader potential of this method/pipeline. The current study demonstrates that a single crystal can yield a complete high-resolution dataset; however, to properly frame the methodology's potential, other lower-symmetry samples should be analyzed and presented.

Therefore, I recommend a major revision to include a broader set of systems, at least two more, that clearly represent a

variety of potential scenarios, such as low-symmetry crystals requiring the collection and merging of multiple datasets, as well as crystals of different morphologies.

Minor concerns:

Before the discussion, the authors state that the level of biologically relevant information obtained by the novel InCellED approach is comparable to that obtained by the well-established SSX. While this may be true, it is also somewhat obvious, its validity will always depend on the resolution limit, and it has already been shown in previous studies using microED. I suggest rephrasing this point to emphasize instead that InCellED provides an additional tool in the methodological toolbox, one that could be advantageous in situations where other techniques are more demanding.

Also, in line with my earlier comments, the discussion includes the statement: "In this study, we established and validated InCellED, a workflow for high-resolution structure determination by 3D ED using only a single intracellular protein crystal in the micrometer size range." The present results only demonstrate that the approach is possible. To substantiate the claim that it constitutes a workflow pipeline, it would need to be shown that it can be successfully applied to multiple systems.

Reviewer #2

(Remarks to the Author)

The manuscript titled "In situ protein crystallography: a single-crystal electron diffraction pipeline for structure determination inside living cells" by Bílá, Pinkas, Khakurel et al. presents a novel and technically significant approach for determining protein structures within living cells using micro-electron diffraction (microED). This work represents a valuable contribution not only to the field of in vivo crystallization, but also to the broader electron diffraction and structural biology communities. This robust in situ approach offers a powerful platform for advancing structural studies of proteins in their native cellular milieu.

That said, the study is innovative and of high potential impact. While the manuscript is generally well written and the methodology is clearly described, I recommend a number of minor revisions to further strengthen its clarity and presentation. My comments and suggestions listed below are intended to support the authors in refining the manuscript and ensuring its readiness for further publication in Nature Communications.

Introduction

The referee recommends including a graphical abstract in the Introduction section to enhance clarity and accessibility. If possible, the authors may consider repositioning Figure 2 as Figure 1 and integrating it directly into this section to provide an early visual overview of the workflow or methodology.

Results

Intracellular crystallization of MgHEX-1

Line 22: 'The crystallization efficiency and low-micrometre size range position HEX-1 crystals as ideal candidates for InCellED method development'

 The referee recommends that the authors specify the expected size range of the HEX-1 crystals and provide an appropriate citation to support this, referencing prior studies where applicable. Quantitative detail would strengthen this statement and contextualize it within existing literature.

In cellulo crystal preparation for ED experiments

 The referee notes that the novelty of the sample preparation method as described is somewhat limited. Upon further examination of the existing literature, similar sample preparation workflows have already been reported for related systems. The primary difference in this study appears to be the use of entire cells as samples, rather than isolated crystals. Given the close methodological overlap, the referee suggests that this section could be merged with the previous one, to streamline the presentation and avoid redundancy.

Additionally, the authors are encouraged to indicate whether alternative grid types were tested during the sample preparation phase. Comparing results across different grids might improve sample distribution and provide further methodological insight.

Electron diffraction data collection.

The referee is interested to know whether, in addition to the previously reported NcHEX-1 structure, the authors have considered using an AlphaFold-predicted model for molecular replacement. Including such an approach could offer an informative structural comparison between the in situ HEX-1 structure, the previously published NcHEX-1, and the AlphaFold model. If feasible, the authors are encouraged to present RMSD values among the three structures, ideally in a tabulated form or as an additional figure, to quantitatively assess structural similarities and deviations. Furthermore, the referee suggests that the authors address the potential issue of radiation damage, particularly given the use of electron diffraction and the presumed sensitivity of in cellulo samples. A brief discussion—supported by any available data or observations—on how radiation damage was minimized or evaluated during data collection would be helpful. Lastly, the authors are requested to specify the angular wedge (tilt range) used during collection of diffraction data. Since small angular wedges at elevated fluence from multiple crystals can help enhance high-resolution signal.

Serial X-ray diffraction data collection, data processing and structure determination & Discussion

The referee would be particularly interested to know whether the authors have considered, or have an opinion on, the use

of a serial data collection approach in electron diffraction. This strategy would be highly relevant in the context of minimizing radiation damage, especially when using high-fluence exposure on single crystals or cells without rotation.

Incorporating such an approach (e.g., serialED) could offer several advantages, including enhanced data quality and a more systematic evaluation of beam-induced damage. Furthermore, if such datasets are or become available, it would be valuable to perform comparative analyses between different collection modes—for example, 3DED vs. serialED—to quantitatively assess radiation effects. Similarly, comparisons could be made to explore vitrification-related artifacts, analogous to what's studied in ED-SSX workflows.

While the referee understands that implementing this methodology may require additional experiments and specific instrumentation, it would be highly appreciated if the authors could address these possibilities in the Discussion section, and provide a brief outlook or commentary on the feasibility and potential impact of such approaches.

Reviewer #3

(Remarks to the Author)

Reviewer #4

(Remarks to the Author)

The manuscript "In Situ protein crystallography: a single crystal electron diffraction pipeline for structure determination inside living cells," by Bílá et al. provides a thorough description of a pipeline for determining crystal structures of proteins grown inside of cells. This process builds upon the InCell Cryst pipeline previously established for serial X-ray crystallography. The authors use the homologous MgHEX-1 protein to demonstrate the process, whereas the original X-ray work had used NcHEX-1. The critical advancement presented here is that the structure could be determined using a single crystal inside a cell, greatly reducing the number of cells and crystals required. The total illuminated crystalline volume between the MicroED and serial X-ray methods is orders of magnitude different, a clear advantage. Although the 1.8 Å resolution obtained by serial synchrotron radiation was slightly higher than the 1.9 Å from MicroED, this served as an excellent validation of the method's accuracy. It was also demonstrated that single diffraction patterns from these crystals could reach resolutions comparable to those obtained by SSX. I believe this paper serves two functions: it broadens the applicability of MicroED to a challenging set of targets, and it demonstrates a workflow for routinely approaching such targets. As such, I believe this work is of high significance to the community and should be accepted for publication.

However, I believe there are several improvements the authors could make that would make the work more approachable, factually accurate, and appealing to those looking to solve protein structures inside of cells.

1. Context/Editorial

The largest criticism I have of this work is not technical, it is editorial. The authors relegate a foundational earlier work from Yang et al. to a single sentence in the discussion. Since planting a flag as the 'first' has little bearing on the quality of this excellent work, the authors should address the Yang et al. study in the introduction. Their work is about using MicroED to solve protein crystal structures from proteins expressed and crystallized in cellulo. The work by Yang et al. used the same core techniques (cryo-FIB milling and MicroED) to determine the structures of proteins that naturally crystallize in human cells. Critically, Yang et al. also correlated their structural data with cryo-electron tomography (cryo-ET), allowing them to map the crystal's 3D location and interactions within the unperturbed cellular landscape. The "InCellED" pipeline, by contrast, is a pure structure-determination method.

Properly discussing this does not damage the current paper; rather, it strengthens it by placing it in the context of other major advances (e.g., milling viscous LCP to study membrane proteins). It would demonstrate to the reader that these in situ approaches are maturing rapidly and have diverse, powerful applications. The authors could also note that cryo-ET could be readily integrated into their workflow if desired.

2. Emphasis on the Single-Crystal Advantage

I encourage the authors to more forcefully emphasize the profound advantages of their single-crystal approach. Solving a high-quality structure from a single crystal lamella and a mere $\sim 1 \mu\text{m}^3$ of illuminated volume is a monumental achievement, especially when contrasted with the >60,000 crystals required for the SSX experiment. The manuscript could be strengthened by explicitly discussing the significant drawbacks inherent to merging such a vast number of microcrystals. The risks of polymorphism, the averaging of partial occupancies that can obscure important structural details, low hit rates, and the immense challenges in data processing and scaling, etc. Their approach bypasses these issues by providing a "clean" dataset from a single, homogenous crystalline entity. This is easy to process and deal with. This is a truly outstanding result, and I feel its impact is somewhat understated in the current text. A more direct comparison would better highlight the transformative power of this work for the structural biology community.

3. Claims

I would be careful with claims about "living cells." Vitrified cells are preserved in a near-native state but are not metabolically active. The fact that protein crystals are grown in living cells is spectacular, but for technical accuracy during data collection, simply saying 'in cells' or 'in situ' is sufficient.

Claims that the crystal structure is more physiological should be made with care. HEX-1 is a key component of Woronin

bodies in filamentous fungi. Here, these crystals are grown inside insect cells, a heterologous host. A statement in the discussion could address this as a known limitation and a fascinating area for future research, such as comparing structures from different expression systems.

4. Clarity

The abstract reads more like a brief introduction. The results are strong enough to speak for themselves. Focusing the abstract more on the quantitative achievements would be more appealing to a broad audience.

The authors alternate between "3D ED," "MicroED," and "3D ED/MicroED." I do not care what you use, but choosing one and using it consistently throughout the manuscript would improve clarity.

The manuscript could benefit from a brief discussion on generalizability. HEX-1 and its homologues might be a "best-case scenario." Acknowledging that the pipeline's performance on more challenging targets remains an open question would add valuable perspective.

Minor note. I don't have an unawkward way of bringing this up, but 'InCellED' is likely to be heard as 'in cel ED' in English. You might consider an alternative to avoid an unintended association. This is editorial and does not affect my scientific assessment.

I hope the authors find our feedback useful, and I look forward to seeing their work in print.

Best,

-Mike Martynowycz

Version 1:

Reviewer comments:

Reviewer #1

(Remarks to the Author)

I believe the manuscript merits publication in its present form. I have no further comments or suggestions for the authors.

I would also like to respond to the authors by quoting Albert Einstein: "The important thing is not to stop questioning. Curiosity has its own reason for existing."

— Albert Einstein, Life Magazine, May 2, 1955.

Reviewer #2

(Remarks to the Author)

I appreciate the authors' comprehensive and well-integrated revisions. The updated manuscript reflects a clear effort to address all comments raised during the review process. The authors have provided additional data and clarifications where requested, improved the methodological description, and strengthened the interpretation of their results. Importantly, the concerns raised by all reviewers have been carefully considered and are now satisfactorily resolved in the revised version. In light of these substantial improvements, I find the manuscript to be significantly strengthened and scientifically sound. I therefore support its publication in Nature Communications.

Reviewer #3

(Remarks to the Author)

Reviewer #4

(Remarks to the Author)

The authors have addressed my comments. Their edits and responses to both mine and the other reviewer's comments have resulted in a much stronger, easier to read manuscript. I look forward to seeing it in press. -MWM

Reply to reviewers' comments

We are grateful to the referees for their exceptionally thorough and thoughtful comments. In response, we revised several parts of the manuscript. Our replies are inserted in blue into the referees' comments below.

Reviewer #4 noted that the abbreviation “*InCellED*” could have unintended associations in English, and we changed the name to “*IncelluloED*”. This name is used throughout the manuscript and in our responses below.

Reviewer #1:

In this manuscript, Redecke and co-workers present an adaptation of the InCellCryst concept, (i.e. a pipeline based on crystallization directly within living insect cells to produce a sufficient amount of homogeneous microcrystals of a target protein for structure determination by serial X-ray, thereby avoiding the purification step), to electron diffraction (InCellED), a technique that is gaining momentum but is still in its infancy in the context of protein crystallography.

They chose HEX-1 from *Magnaporthe grisea* (MgHEX-1) crystals, grown in insect cells to preserve their native state, to demonstrate that electron diffraction data collected from a single crystal can yield enough information to determine the previously unknown 3D structure at 1.9 Å resolution. They also compare these results with a similar dataset obtained by serial X-ray crystallography, which used tens of thousands of crystals and reached a resolution limit of 1.8 Å. Because electron diffraction requires far fewer cell-containing crystals, this method could help overcome the bottleneck of low efficiency in *in cellulo* crystal growth. Nevertheless, sample production, preparation, and adaptation for 3D microED data collection require specialized know-how, which is still under development.

RESPONSE: This is correct, and our present work contributes directly to building the requisite knowledge base. *IncelluloED* provides a direct means to monitor and analyse intracellular crystallization processes right there where the crystallization occurs. Such studies could offer insights into fundamental aspects, including the mechanisms of *in cellulo* crystallization, the fraction of proteins that crystallize in cells, and the proportion of crystals that diffract. While the methodology requires skilled personnel and specialist knowledge, the *IncelluloED* pipeline is robust, and no crystals were lost during our studies. To emphasize this point, we added an additional comment on page 9 (first paragraph) regarding the performance of the method: “During the development and optimization steps of the *IncelluloED* sample preparation workflow, we used 14 cells containing MgHEX-1 microcrystals, selected by RL/FL imaging. All crystals were successfully milled into lamellae, and all lamellae provided high-resolution ED signals.”

The chosen system, MgHEX-1, is consistent with their previously characterized *Neurospora crassa* HEX-1 (NcHEX-1), since both naturally self-assemble into a hexagonal crystalline material. The new MgHEX-1 structure could shed light on the mechanism underlying the self-assembly process that drives cellular stress but further analysis seems to be needed to find the mechanism behind its formation naturally.

The procedure involves cryogenic focused ion beam milling (cryo-FIB milling) coupled to scanning electron microscopy (SEM) to adjust sample thickness after locating crystals within the cell using fluorescence (FL) microscopy, as well as cryo-light microscopy and correlative light–electron microscopy. The sample preparation process is well described; however, fine details (beyond the scope of the present work) are not provided, and reproducibility may be challenging even for experienced technicians.

RESPONSE: While the methodology requires skilled personnel and specialist knowledge, the *IncelluloED* pipeline is robust, and we lost no crystals during our studies. Following the advice of Referee #2, we introduced a graphical abstract in the Introduction section (new Figure 1) to enhance clarity and accessibility. We also expanded and restructured the *Results* section with a detailed description of the procedures under the heading “Workflow of *in cellulo* crystal preparation for ED experiments” on pages 6-7 to highlight the key steps of sample preparation and the expected results. The *Methods* section was also restructured accordingly (pages 17-21). We believe these changes will facilitate the adoption of the *IncelluloED* method by new users.

Overall, the manuscript is well written, and the methodology, its main focus, is clearly explained, even from the perspective of a non-expert in electron diffraction. Therefore, I believe this article will be significant for the structural biology community, although I have one major concern regarding its framing.

Major concerns:

Although this approach for InCelleD clearly has high potential and could become a viable alternative technique, the data presented in the manuscript do not allow one to envision its general applicability.

Looking at the authors’ own previous work in this journal, it is evident that they have several other systems available that could be used to showcase the broader potential of this method/pipeline. The current study demonstrates that a single crystal can yield a complete high-resolution dataset; however, to properly frame the methodology’s potential, other lower-symmetry samples should be analyzed and presented.

Therefore, I recommend a major revision to include a broader set of systems, at least two more, that clearly represent a variety of potential scenarios, such as low-symmetry crystals requiring the collection and merging of multiple datasets, as well as crystals of different morphologies.

RESPONSE: The aim of this paper is to demonstrate that it is possible to obtain a high-resolution structure from a single protein crystal within a single cell, using the new *IncelluloED* pipeline. Our results show that *in cellulo* electron crystallography can be performed many orders of magnitude more efficiently than synchrotron serial X-ray crystallography with the help of the *IncelluloED* pipeline. For quantitative data on efficiency see the end of para 3, page 12 and Supplementary Text 1.

We agree with Reviewers #1 and #4 that the generalisability and limitations of the method should be discussed more explicitly. In response, we have substantially revised the *Discussion* section. On page 12, para 2 we discuss heterologous protein expression, a point raised by Referee #4. On page 13, para 1, we describe data collection strategies for low symmetry crystals (point raised by Referee #1). We explain that “For all space groups except triclinic P1, a 90° wedge can achieve >90% completeness, desired for reliable structure

refinement and model building⁶⁰, provided the crystal is favourably oriented⁵⁹.” We discuss options to improve completeness either by manually rotating the frozen sample grid in the cryo-FIB/SEM holder prior to milling to bring crystals into a favorable orientation based on their morphology and orientation on the frozen grid (which can be observed by cryo-light microscopy), or by using a dual-axis (double-tilt) cryo-holder. P1 crystals are special and require 180° rotation of the sample grid, and this is not possible in cryo-TEMs due to practical constraints like increased lamella thickness at high tilts and obstruction by grid bars (Supplementary Figure 2), which restrict the usable tilt range to about 110°–120°, despite nominal hardware limits of –70° to +70°. If high completeness cannot be achieved easily, data from multiple crystals can be merged, as is common practice in X-ray and electron crystallography. All these points are incorporated into the revised *Discussion* section with appropriate references.

Minor concerns:

Before the discussion, the authors state that the level of biologically relevant information obtained by the novel InCellIED approach is comparable to that obtained by the well-established SSX. While this may be true, it is also somewhat obvious, its validity will always depend on the resolution limit, and it has already been shown in previous studies using microED. I suggest rephrasing this point to emphasize instead that InCellIED provides an additional tool in the methodological toolbox, one that could be advantageous in situations where other techniques are more demanding.

RESPONSE: We introduced the following sentence on page 11, para 3 to address this concern: “*IncelluloED* is an additional tool in the methodological toolbox, one that could be advantageous in situations where other techniques are more demanding.”

Also, in line with my earlier comments, the discussion includes the statement: “In this study, we established and validated InCellIED, a workflow for high-resolution structure determination by 3D ED using only a single intracellular protein crystal in the micrometer size range.” The present results only demonstrate that the approach is possible. To substantiate the claim that it constitutes a workflow pipeline, it would need to be shown that it can be successfully applied to multiple systems.

RESPONSE: We agree. We only demonstrate here that the approach is possible and produces excellent results on MgHEX-1 crystals in insect cells. Para 1 of the newly rewritten *Discussion* section states “We introduced and validated the *IncelluloED* workflow for high-resolution structure determination of intracellularly crystallized recombinant proteins, using 3D electron diffraction from a single crystal inside a single cell. As a proof of concept, we determined the previously unknown structure of the HEX-1 protein from *Magnaporthe grisea*, selected as a pilot target. The *IncelluloED* pipeline achieved comparable resolution and identical structural information to serial synchrotron X-ray crystallography (SSX) while reducing the required sample volume by seven orders of magnitude to about 1 μm³, and potentially less. Its performance on other targets remains to be established, and experimental validation across a broader range of proteins will be the focus of future work.” (page 12)

In this context, it may perhaps be appropriate - though excessive - to invoke Newton’s advice: “To explain all nature is too difficult a task for any one man or even for any one age. ’Tis much better to do a little with

certainty & leave the rest for others that come after than to explain all things by conjecture without making sure of any thing.” (Isaac Newton, *The Correspondence of Isaac Newton*. Edited by H. W. Turnbull, vol. 5, Cambridge University Press, 1961. Draft of a letter to Roger Cotes, June 1712).

Reviewer #2:

The manuscript titled “In situ protein crystallography: a single-crystal electron diffraction pipeline for structure determination inside living cells” by Bílá, Pinkas, Khakurel et al. presents a novel and technically significant approach for determining protein structures within living cells using micro-electron diffraction (microED). This work represents a valuable contribution not only to the field of in vivo crystallization, but also to the broader electron diffraction and structural biology communities. This robust in situ approach offers a powerful platform for advancing structural studies of proteins in their native cellular milieu.

That said, the study is innovative and of high potential impact. While the manuscript is generally well written and the methodology is clearly described, I recommend a number of minor revisions to further strengthen its clarity and presentation. My comments and suggestions listed below are intended to support the authors in refining the manuscript and ensuring its readiness for further publication in Nature Communications.

Introduction

The referee recommends including a graphical abstract in the Introduction section to enhance clarity and accessibility. If possible, the authors may consider repositioning Figure 2 as Figure 1 and integrating it directly into this section to provide an early visual overview of the workflow or methodology.

RESPONSE: We fully agree, and we added a graphical abstract to the *Introduction* section (new Figure 1) to improve clarity and accessibility. We also expanded and reorganised the *Results* section, adding a detailed description of the procedures: “Workflow of *in cellulo* crystal preparation for ED experiments” (pages 6–7) to highlight the key steps of sample preparation and the expected outcomes. The *Methods* section has also been restructured accordingly (see the very detailed subsection on “*In cellulo* crystal preparation for ED experiments” on pages 17-21). We believe these changes will facilitate the adoption of the *IncelluloED* method by new users.

Results

Intracellular crystallization of MgHEX-1

Line 22: ‘The crystallization efficiency and low-micrometre size range position HEX-1 crystals as ideal candidates for InCelleD method development’

 The referee recommends that the authors specify the expected size range of the HEX-1 crystals and provide an appropriate citation to support this, referencing prior studies where applicable. Quantitative detail would strengthen this statement and contextualize it within existing literature.

RESPONSE: We agree and we added the requisite information on page 5 (first paragraph) including appropriate citations: “A *Neurospora crassa* HEX-1 (*NcHEX-1*) was found to produce regular, hexagonal

crystals with average dimensions of $9.1 \pm 3.2 \mu\text{m}$ (\pm standard deviation) in length and $3.5 \pm 0.7 \mu\text{m}$ in width when overexpressed in insect cells using the recombinant baculovirus (rBV) system^{8,38}.”

In cellulo crystal preparation for ED experiments

 The referee notes that the novelty of the sample preparation method as described is somewhat limited. Upon further examination of the existing literature, similar sample preparation workflows have already been reported for related systems. The primary difference in this study appears to be the use of entire cells as samples, rather than isolated crystals. Given the close methodological overlap, the referee suggests that this section could be merged with the previous one, to streamline the presentation and avoid redundancy.

RESPONSE: As mentioned above, we rewrote the section “Workflow of *in cellulo* crystal preparation for ED experiments” (pages 6-7) in the *Results* section in a way that its subsections follow the schematic overview in the new Figure 1. In addition, there is now a subsection “Cell collection and freezing on TEM grids” (page 6) which is introduced to address the point of the referee.

Additionally, the authors are encouraged to indicate whether alternative grid types were tested during the sample preparation phase. Comparing results across different grids might improve sample distribution and provide further methodological insight.

RESPONSE: There were no systematic comparisons of different grid types. We reported the observation in the section “*In cellulo* crystal preparation for ED experiments” of *Methods* that holey gold-film grids (UltrAuFoil® R2/2, 200 gold mesh, Quantifoil Micro Tools) performed well, and the gold film appeared to be more sturdy than the holey carbon film (QUANTIFOIL®, Quantifoil Micro Tools), and we continued using the gold grids. To avoid implying generality, we added the phrase “in our hands” to the relevant sentence (page 18, para 1) to indicate that this is a practical advice rather than a systematic conclusion.

Electron diffraction data collection.

The referee is interested to know whether, in addition to the previously reported NcHEX-1 structure, the authors have considered using an AlphaFold-predicted model for molecular replacement. Including such an approach could offer an informative structural comparison between the *in situ* HEX-1 structure, the previously published NcHEX-1, and the AlphaFold model. If feasible, the authors are encouraged to present RMSD values among the three structures, ideally in a tabulated form or as an additional figure, to quantitatively assess structural similarities and deviations.

RESPONSE: This is an interesting suggestion and yes, we calculated an Alphafold 3 prediction for the MgHEX-1 structure and used it for molecular replacement trials, and it worked well. However, it turned out that the subsequent refinement went faster in our hands when we used the experimental NcHEX-1 (PDB: 1KHI) structure as a search model. The structural comparison of the predicted MgHEX-1 AlphaFold model and our experimental 3D ED microvolume and the serial X-ray structures shows highly comparable protein folds with only minor deviations, characterized by RMSD values of in the range of 1 Å. This is expected, because HEX-1 proteins are comparatively small (molecular mass around 20 kDa), and they are well-folded proteins closely related to the NcHEX-1 structure, already deposited in PDB and thus part of the training data set for AlphaFold 3.

However, since the AlphaFold comparison might be interesting for the readership, we included the structural comparison of our experimental *Mg*HEX-1 structures with the AlphaFold 3 model in the new Supplementary Figure 10. The requested RMSD values are now presented in the legend of this figure: “The C α atoms of the AlphaFold 3 model show an average RMSD of 1.17 Å and of 1.09 Å relative to the equivalent atoms of the 3D ED microvolume and the SSX structure, respectively, indicating no significant structural differences.” We also added a sentence to *Results* (page 10, para 2): “Similar RMSD values (about 1 Å) were obtained by comparing the Ca atom positions of the 3D ED microvolume and the SSX structures with that of a predicted AlphaFold 3 model⁴⁹ for *Mg*HEX-1 (Supplementary Fig. 10).”

Furthermore, the referee suggests that the authors address the potential issue of radiation damage, particularly given the use of electron diffraction and the presumed sensitivity of in cellulo samples. A brief discussion—supported by any available data or observations—on how radiation damage was minimized or evaluated during data collection would be helpful.

RESPONSE: We have added extensive information on radiation damage to the paper. This includes quantitative data in *Results* on pages 7-8 coupled with new Supplementary Figure 8 and its legend, and description of radiation damage experiments in *Methods* (first paragraph in the section of “ED data collection, data processing and structure determination”; page 21). The new *Discussion* section gives further information on radiation damage (page 13, para 2).

Lastly, the authors are requested to specify the angular wedge (tilt range) used during collection of diffraction data. Since small angular wedges at elevated fluence from multiple crystals can help enhance high-resolution signal.

RESPONSE: We specified the tilt range in Table 1.

Serial X-ray diffraction data collection, data processing and structure determination & Discussion

The referee would be particularly interested to know whether the authors have considered, or have an opinion on, the use of a serial data collection approach in electron diffraction. This strategy would be highly relevant in the context of minimizing radiation damage, especially when using high-fluence exposure on single crystals or cells without rotation.

Incorporating such an approach (e.g., serialED) could offer several advantages, including enhanced data quality and a more systematic evaluation of beam-induced damage. Furthermore, if such datasets are or become available, it would be valuable to perform comparative analyses between different collection modes—for example, 3DED vs. serialED—to quantitatively assess radiation effects. Similarly, comparisons could be made to explore vitrification-related artifacts, analogous to what's studied in ED-SSX workflows.

While the referee understands that implementing this methodology may require additional experiments and specific instrumentation, it would be highly appreciated if the authors could address these possibilities in the Discussion section, and provide a brief outlook or commentary on the feasibility and potential impact of such approaches.

RESPONSE: Serial protein crystallography with electron diffraction is not new, it was first introduced as “SerialED” by Bücker *et al.* [*Nat Commun* 11, 996 (2020), <https://doi.org/10.1038/s41467-020-14793-0>]. A recent *bioRxiv* preprint by Gallagher-Jones *et al.* [*bioRxiv* (2025), <https://doi.org/10.1101/2025.06.24.661318>] shows that high-resolution protein structures can be obtained using more than ten thousand crystals grown *in cellulo* and then extracted out of the cells for data collection in a cryo-TEM. Serial crystallography gives structures with extremely low-doses (page 12, paras 1 and 3). The main problem is that the efficiency of intracellular crystallization can be very low (frequently less than 0.1%) and this limits the use of serial crystallography with intracellularly grown crystals in general. Another limiting factor is that during the extraction process, conditions change and small-molecule ligands may diffuse out of microcrystals. Merging data from thousands of crystals can obscure heterogeneity, mask twinning, impose artificial symmetry, and smoothen out conformational variations, dynamics, and rare states (page 12, paragraph 3). For SerialED, crystals must be sufficiently thin and the grids well blotted to ensure adequate electron transmission during cryo-TEM ED experiments. In contrast, *IncelluloED* allows optimization of crystal thickness through cryo-FIB milling and provides high-resolution structures with the same level of structural detail as serial synchrotron crystallography, while using a seven-million-fold smaller sample volume and a remarkably low dose. An $\sim 80^\circ$ data wedge from a MgHEX-1 crystal required only ~ 1.1 MGy for a structure at 1.9 Å resolution, and this is far from the generally observed dose limits for high-resolution studies by electron diffraction (Supplementary Figure 8; Table 1; new *Discussion* on page 13, para 2 with refs. therein).

Single-crystal diffraction methods such as *IncelluloED* offer additional possibilities for radiation-damage control and dose reduction, compared with serial crystallography. Instead of using a large number of crystals and taking still or small-rotation exposures, a strategy based on an idea described in Berglund *et al.* [*Nature* 417, 463–468 (2002), <https://doi.org/10.1038/417463a>] uses 3D data sets from a small number of crystals, about a dozen or so. This number is in the ball park of the daily production capabilities of cryo-FIB lamella milling with current technologies (page 14, para 2). Data collection from each lamella by electron diffraction would start at systematically offset crystal orientations as illustrated in Figure 3a of the Berglund *et al.* paper, and proceed until full 3D data sets are recorded from all lamellae. Note that the method also works with a somewhat larger number of randomly oriented crystals. By selecting and combining defined sections of each 3D data set, the method redistributes the absorbed dose to create a number of “composite 3D data sets,” from which a number of “composite structures” can be calculated each at a different dose level. At the low-dose end, 3D structures with doses in the kGy range can be obtained, while the high-dose end shows structures with several MGy doses. Each “composite structure” represents a frame from a damage movie, showing radiation-induced changes in the structure as a function of the absorbed dose.

The method is flexible, and key parameters can be changed. This includes choosing the number of lamellae to be used, the total dose a lamella receives, and the angular range used for collecting the individual data sets. By reducing the angular range, “small-wedge” data acquisition can be done, and by going further, a series of still diffraction patterns can be recorded. A fascinating aspect of this method is that most of these operations can be performed computationally after data collection, provided that enough data are available. For instance, posthumous resolution extension is possible from the 3D data sets by selecting and combining those data wedges where the signal-to-noise ratios are the highest, the relative diffraction intensity changes the smallest, the unit cell dimensions remain unchanged, and data statistic parameters such as $CC_{1/2}$ and CC^* are optimal.

All of the above is concisely addressed in the new *Discussion* on pages 12-14.

Reviewer #3:

Remarks to the Author:

The authors' exploration of an alternative method, MicroED/3D ED, for the direct analysis of protein crystals grown in cells represents a fascinating and valuable contribution. I believe that the MicroED approach is particularly advantageous in cases where obtaining large protein crystals is challenging. The authors have also appropriately discussed the limitations of their study and proposed possible directions for future work.

One question for consideration: are these crystals separable from the cells? If so, such separation might enable future crystallization through seeding, which could be beneficial in cases where cell preparation is not feasible for repetitive processes.

RESPONSE: We thank the referee for this suggestion. It depends on the individual protein if intracellular crystals can be isolated from cells. Some proteins form highly stable crystals, others form crystals that immediately dissolve after cell lysis. *MgHEX-1* crystals can be isolated, but a change of the environment might impact the crystal quality, even if it is not directly visible. Thus, we highly recommend not to isolate crystals that have been grown in a cell for diffraction. However, we agree that seeding with isolated fragments of these crystals can be beneficial for conventional crystallization of proteins *in vitro*. Such experiments are currently ongoing.

However, I believe the manuscript would benefit from addressing the following points:

1. Abstract focus – The abstract primarily emphasizes the importance of this work and its contribution to the field of in-cell protein crystallography, rather than highlighting the key findings and specific results of this study. Consider revising to give more weight to the main achievements.

RESPONSE: We agree. We revised the abstract accordingly, also considering the suggestions of Reviewer #4.

2. Abbreviations and materials – The large number of abbreviations and parenthetical material sources within the main text disrupts the reading flow. Moving abbreviations to a dedicated “Abbreviations” section and listing material sources in the “Methods” section would improve clarity.

RESPONSE: We have substantially revised the *Introduction* and *Discussion*, and improved other sections as well, using abbreviations less frequently to enhance readability and flow. We believe these changes improve the clarity of the manuscript. We would prefer to leave the decision on including an abbreviation list and

detailed source listings to the editor, as this practice does not appear to be consistent with most papers published in the journal.

3. Clarifications in results:

o *Line 102*: “MgHEX-1” appears to be the protein code but is placed directly after the fungus name *Magnaporthe grisea*. This placement may be confusing.
o *Line 114*: In the results section, “MgHEX-1” is placed next to the approach name— please confirm this usage is correct.

RESPONSE: MgHEX-1 is indeed the abbreviation for “HEX-1 protein from [the filamentous fungus] *Magnaporthe grisea*”. Thus, the placement of the abbreviation (MgHEX-1) right after the name of the fungus is correct in our opinion. On page 5, para 1, the following sentence “To establish the *InCelluloED* approach, MgHEX-1, a structurally uncharacterized HEX-1 protein derived from the filamentous fungus *M. grisea*, has been selected.” is correct, since the approach name and the protein abbreviation are separated by a comma.

o *Line 127*: The text states that three crystals per cell were identified; however, Supplementary Fig. 1 shows only one crystal per cell. Please reconcile this discrepancy.

RESPONSE: Text from page 5, para 2: “Applying the *InCellCryst* approach, UP TO three hexagonal-shaped structures per cell were detected by light microscopy...”. This means that we have cells with none, one, two, or three crystals inside. We have not seen more. See also Figure 2 (previously Figure 1).

4. Line 177 – The sentence “ED datasets were characterized by highly similar data processing statistics, and one of them was selected...” lacks details on what statistical criteria were used. A reference or specific values would improve clarity.

RESPONSE: We agree with the referee. To address this, we included the text “These two microvolume ED data sets gave very similar data processing statistics (**Supplementary Table 1**).” in page 8, para 2, and the new Supplementary Table 1 was added into Supplementary Information.

5. Lines 175 and 185 – Two different fluences (0.3 and 0.6 e⁻/Å²) were used for similar experiments involving different crystal volumes. The rationale for selecting these different fluences should be explained.

RESPONSE: We added the following sentence on page 8, para 3: “Three-dimensional ED data were collected from one lamella at a total fluence of ~0.6 e⁻/Å², which was double of the fluence used for the microvolume ED data collection to compensate for the reduced sample volume.”

6. Line 224 – RMSD values are mentioned here but are not referenced in the “3D ED Structure” section. Consider cross-referencing for consistency.

RESPONSE: We included the RMSD values of the 3D ED structures to the MR search model in the corresponding sentence at page 10, para 1: “The calculated RMSD of 0.520 Å compared to the 134 equivalent C α atoms of the MR search model is increased compared to that of 0.341 Å and 0.340 Å calculated for the

micro- and nano-volume 3D ED structures.” These values were already presented in the “3D ED structure of MgHEX-1” section of the *Results*.

7. Sentence length – Some sentences, such as the one on line 224, are overly long and difficult to follow. Breaking them into shorter sentences would improve readability.

RESPONSE: We agree that some sentences might be a bit too long and thus difficult to understand. We revised the manuscript in this context. For example, we split the sentence mentioned by the reviewer into two sentences.

8. Gold vs. carbon film grids – Line 406: When stating that gold film grids have better mechanical stability, it would be valuable to note whether any comparative data were collected using carbon film grids and, if so, whether any differences in data quality were observed.

RESPONSE: There were no systematic comparisons of different grid types. The holey gold-film grids (UltraAuFoil® R2/2, 200 gold mesh, Quantifoil Micro Tools) performed well in our hands and the gold film appeared to be more sturdy than the holey carbon film (QUANTIFOIL®, Quantifoil Micro Tools), and we so continued using the gold grids (page 18, para 1). See also reply to Referee #2 about the same subject.

9. Lines 410 and 412 – The expressions “keeping the sample at <-170°C” and “while being kept at <-153°C” should be revised to “kept below” or “maintained below” for clarity and grammatical correctness.

RESPONSE: We incorporated the changes as suggested.

Overall, this is a strong piece of work with significant potential impact, and addressing these points would further strengthen the manuscript’s clarity and scientific rigor.

Reviewer #4:

The manuscript “In Situ protein crystallography: a single crystal electron diffraction pipeline for structure determination inside living cells,” by Bilá et al. provides a thorough description of a pipeline for determining crystal structures of proteins grown inside of cells. This process builds upon the InCell Cryst pipeline previously established for serial X-ray crystallography. The authors use the homologous MgHEX-1 protein to demonstrate the process, whereas the original X-ray work had used NcHEX-1. The critical advancement presented here is that the structure could be determined using a single crystal inside a cell, greatly reducing the number of cells and crystals required. The total illuminated crystalline volume between the MicroED and serial X-ray methods is orders of magnitude different, a clear advantage. Although the 1.8 Å resolution obtained by serial synchrotron radiation was slightly higher than the 1.9 Å from MicroED, this served as an excellent validation of the method's accuracy. It was also demonstrated that single diffraction patterns from these crystals could reach resolutions comparable to those obtained by SSX. I believe this paper serves two functions: it broadens the applicability of MicroED to a challenging set of targets, and it demonstrates a

workflow for routinely approaching such targets. As such, I believe this work is of high significance to the community and should be accepted for publication.

However, I believe there are several improvements the authors could make that would make the work more approachable, factually accurate, and appealing to those looking to solve protein structures inside of cells.

1. Context/Editorial

The largest criticism I have of this work is not technical, it is editorial. The authors relegate a foundational earlier work from Yang et al. to a single sentence in the discussion. Since planting a flag as the ‘first’ has little bearing on the quality of this excellent work, the authors should address the Yang et al. study in the introduction. Their work is about using MicroED to solve protein crystal structures from proteins expressed and crystallized in cellulo. The work by Yang et al. used the same core techniques (cryo-FIB milling and MicroED) to determine the structures of proteins that naturally crystallize in human cells. Critically, Yang et al. also correlated their structural data with cryo-electron tomography (cryo-ET), allowing them to map the crystal's 3D location and interactions within the unperturbed cellular landscape. The "InCelleD" pipeline, by contrast, is a pure structure-determination method.

Properly discussing this does not damage the current paper; rather, it strengthens it by placing it in the context of other major advances (e.g., milling viscous LCP to study membrane proteins). It would demonstrate to the reader that these *in situ* approaches are maturing rapidly and have diverse, powerful applications. The authors could also note that cryo-ET could be readily integrated into their workflow if desired.

RESPONSE: We agree with the referee. To address the points, we have significantly revised the whole *Introduction*. We added a paragraph on page 4 to highlight the work of Yang *et al.* [*In situ* crystalline structure of the human eosinophil major basic protein-1. *bioRxiv* (2024), <https://doi.org/10.1101/2024.10.09.617336> - **ref. #25**]: “The possibility of collecting ED data sets from multiple intracellular crystals in frozen cells was recently demonstrated for the major basic protein-1 (MBP-1)²⁵, which naturally forms numerous nanocrystals in the cytoplasmic secretory granules of eosinophils. The study also showed that combining cryo-electron tomography²⁶ with 3D ED enables analysis of physiologically relevant structural changes associated with the release of MBP-1 *in situ* and on multiple length scales, from molecular structure to nanometer-scale changes in the crystal packing during MBP-1 release.”

Milling viscous lipidic cubic phase (LCP) samples to study membrane proteins was first introduced by some of the authors of the present manuscript [Polovinkin *et al.* Demonstration of electron diffraction from membrane protein crystals grown in a lipidic mesophase after lamella preparation by focused ion beam milling at cryogenic temperatures. *J. Appl. Cryst.* 53, 1416–1424 (2020); <https://doi.org/10.1107/S1600576720013096>]. This work served as an important stimulus for the development of the *IncelluloED* concept. We cite this paper [**ref. #31**] on page 6, paragraph 1: "Building upon the development of cryo-FIB milling procedures for crystals^{31,33,41} and also on the idea of using 3D negative fluorescence imaging for accurate 3D localization of target crystals inside a cryo-FIB/SEM microscope, we established the protocol for ED sample preparation. "

2. Emphasis on the Single-Crystal Advantage

I encourage the authors to more forcefully emphasize the profound advantages of their single-crystal approach.

Solving a high-quality structure from a single crystal lamella and a mere $\sim 1\mu\text{m}^3$ of illuminated volume is a monumental achievement, especially when contrasted with the $>60,000$ crystals required for the SSX experiment. The manuscript could be strengthened by explicitly discussing the significant drawbacks inherent to merging such a vast number of microcrystals. The risks of polymorphism, the averaging of partial occupancies that can obscure important structural details, low hit rates, and the immense challenges in data processing and scaling, etc. Their approach bypasses these issues by providing a "clean" dataset from a single, homogenous crystalline entity. This is easy to process and deal with. This is a truly outstanding result, and I feel its impact is somewhat understated in the current text. A more direct comparison would better highlight the transformative power of this work for the structural biology community.

RESPONSE: We absolutely agree with the referee. We have revised the entire *Discussion* to underline the single-crystal "advantage" of the *IncelluloED* approach and to address related points from other reviewers, while also ensuring the *Discussion* section remains concise.

3. Claims

I would be careful with claims about "living cells." Vitrified cells are preserved in a near-native state but are not metabolically active. The fact that protein crystals are grown in living cells is spectacular, but for technical accuracy during data collection, simply saying 'in cells' or 'in situ' is sufficient.

RESPONSE: The referee is right. We were using cells that were alive at the point of freezing for lamella preparation. We changed the text as suggested by the referee. We also changed the title of the manuscript from "*In situ* protein crystallography: a single-crystal electron diffraction pipeline for structure determination inside living cells" to "*In situ* protein crystallography: a single-crystal electron diffraction pipeline for direct structure determination inside cells".

Claims that the crystal structure is more physiological should be made with care.

RESPONSE: We agree that claims that a structure from an intracellular crystal is physiological should be made with care. However, our manuscript does not contain such a claim. We state at the beginning of the *Introduction* (first paragraph): "In certain cases, the quasi-native cellular environment enables binding and identification of physiologically relevant ligands and cofactors^{2,3}, which AI-based structure prediction cannot yet replicate." This is a statement that refers to our previous work, where we could clearly show that cofactors present in the cellular environment are bound to the enzymes and are visible in the electron density maps after structure determination. And since the native environment of a cell contains the entire set of cofactors, there is a selection for the one with highest binding affinity, which is usually the physiologically relevant cofactor or inhibitor. We do not state that the crystal structure is more physiological, only that the cellular environment provides the opportunity for cofactor selection and binding at near-physiological effector concentrations.

HEX-1 is a key component of Woronin bodies in filamentous fungi. Here, these crystals are grown inside insect cells, a heterologous host. A statement in the discussion could address this as a known limitation and a fascinating area for future research, such as comparing structures from different expression systems.

RESPONSE: We incorporated the suggestion into *Discussion* on page 12, para 2: “A key question is how broadly applicable *IncelluloED* will be. We studied crystals of a fungal protein grown inside an insect cell, a heterologous host to HEX-1. Heterologous expression can present challenges, but an analysis of proteins in the PDB with identical sequences produced in native versus recombinant hosts found no cases where the same sequence adopted a significantly different global fold⁵⁰. Post-translational modifications, host-specific chaperones, or the assembly of multi-component complexes may affect applicability and will need to be explored.”

4. Clarity

The abstract reads more like a brief introduction. The results are strong enough to speak for themselves. Focusing the abstract more on the quantitative achievements would be more appealing to a broad audience.

RESPONSE: We agree and improved the abstract accordingly.

ABSTRACT: Intracellular crystallisation is an emerging approach in structural biology that bypasses the need for protein purification. In 2024, the *InCellCryst* pipeline was introduced for structural studies of intracellular crystals by serial X-ray crystallography. Serial crystallography requires the exposure of tens of thousands of cells containing intracellular crystals, precluding high-resolution studies on proteins that crystallize only in a few cells. Here we introduce *IncelluloED*, a method that combines intracellular crystallization with *in situ* 3D electron diffraction in cells and achieves high-resolution structures from just one crystal inside one cell. Experiments on a microcrystal of the HEX-1 protein from *Magnaporthe grisea*, grown inside an insect cell, gave a structure at 1.9 Å resolution from a volume of ~1.6 μm³ as compared to 1.8 Å resolution achieved by serial X-ray crystallography from a combined volume exceeding eleven million μm³. *IncelluloED* uses widely available cryo-EM tools and brings high-resolution structural biology into home laboratories while also advancing a vision for a “single-cell structural laboratory”.

The authors alternate between "3D ED," "MicroED," and "3D ED/MicroED." I do not care what you use, but choosing one and using it consistently throughout the manuscript would improve clarity.

RESPONSE: We agree with the referee, and we are now using 3D ED throughout the manuscript.

The manuscript could benefit from a brief discussion on generalizability. HEX-1 and its homologues might be a "best-case scenario." Acknowledging that the pipeline's performance on more challenging targets remains an open question would add valuable perspective.

RESPONSE: We agree with the referee. This comment is similar to one from Reviewer #1. The significantly revised *Discussion* now addresses “generalizability”. We also ensured that the *Discussion* section was concise. We acknowledge “that the pipeline's performance on more challenging targets remains an open question” (page 12, para 1): “Its performance on other targets remains to be established, and experimental validation across a broader range of proteins will be the focus of future work.”

Minor note. I don't have an awkward way of bringing this up, but 'InCelleD' is likely to be heard as 'inced' in English. You might consider an alternative to avoid an unintended association. This is editorial and does not affect my scientific assessment.

RESPONSE: We thank the referee for pointing this out and changed the name of the pipeline from *InCelleD* to *IncelluloED*.

I hope the authors find our feedback useful, and I look forward to seeing their work in print.

Reply to reviewers' comments

We are grateful to the four reviewers for their exceptionally useful and insightful comments, which significantly strengthened the paper.

Reviewer #1:

I believe the manuscript merits publication in its present form. I have no further comments or suggestions for the authors.

I would also like to respond to the authors by quoting Albert Einstein: “The important thing is not to stop questioning. Curiosity has its own reason for existing.” — Albert Einstein, Life Magazine, May 2, 1955.

Reviewer #2:

I appreciate the authors' comprehensive and well-integrated revisions. The updated manuscript reflects a clear effort to address all comments raised during the review process. The authors have provided additional data and clarifications where requested, improved the methodological description, and strengthened the interpretation of their results. Importantly, the concerns raised by all reviewers have been carefully considered and are now satisfactorily resolved in the revised version.

In light of these substantial improvements, I find the manuscript to be significantly strengthened and scientifically sound. I therefore support its publication in Nature Communications.

Reviewer #3:

Remarks to the Author:

The authors have satisfactorily addressed all of my questions and concerns and have revised the manuscript appropriately. Their exploration of MicroED/3D ED as an alternative approach for the direct analysis of protein crystals grown in cells constitutes a significant and intriguing contribution to the field.

Reviewer #4:

The authors have addressed my comments. Their edits and responses to both mine and the other reviewer's comments have resulted in a much stronger, easier to read manuscript. I look forward to seeing it in press. -
MWM

1. Remarks to the Author:

The authors' exploration of an alternative method, MicroED/3D ED, for the direct analysis of protein crystals grown in cells represents a fascinating and valuable contribution. I believe that the MicroED approach is particularly advantageous in cases where obtaining large protein crystals is challenging. The authors have also appropriately discussed the limitations of their study and proposed possible directions for future work.

One question for consideration: are these crystals separable from the cells? If so, such separation might enable future crystallization through seeding, which could be beneficial in cases where cell preparation is not feasible for repetitive processes.

However, I believe the manuscript would benefit from addressing the following points:

1. **Abstract focus** – The abstract primarily emphasizes the importance of this work and its contribution to the field of in-cell protein crystallography, rather than highlighting the key findings and specific results of this study. Consider revising to give more weight to the main achievements.
2. **Abbreviations and materials** – The large number of abbreviations and parenthetical material sources within the main text disrupts the reading flow. Moving abbreviations to a dedicated “Abbreviations” section and listing material sources in the “Methods” section would improve clarity.
3. **Clarifications in results:**
 - *Line 102:* “MgHEX-1” appears to be the protein code but is placed directly after the fungus name *Magnaporthe grisea*. This placement may be confusing.
 - *Line 114:* In the results section, “MgHEX-1” is placed next to the approach name— please confirm this usage is correct.
 - *Line 127:* The text states that three crystals per cell were identified; however, Supplementary Fig. 1 shows only one crystal per cell. Please reconcile this discrepancy.
4. **Line 177** – The sentence “ED datasets were characterized by highly similar data processing statistics, and one of them was selected...” lacks details on what statistical criteria were used. A reference or specific values would improve clarity.
5. **Lines 175 and 185** – Two different fluences (0.3 and $0.6 \text{ e}^-/\text{\AA}^2$) were used for similar experiments involving different crystal volumes. The rationale for selecting these different fluences should be explained.
6. **Line 224** – RMSD values are mentioned here but are not referenced in the “3D ED Structure” section. Consider cross-referencing for consistency.
7. **Sentence length** – Some sentences, such as the one on line 224, are overly long and difficult to follow. Breaking them into shorter sentences would improve readability.

8. **Gold vs. carbon film grids** – Line 406: When stating that gold film grids have better mechanical stability, it would be valuable to note whether any comparative data were collected using carbon film grids and, if so, whether any differences in data quality were observed.
9. **Lines 410 and 412** – The expressions “keeping the sample at $<-170^{\circ}\text{C}$ ” and “while being kept at $<-153^{\circ}\text{C}$ ” should be revised to “kept below” or “maintained below” for clarity and grammatical correctness.

Overall, this is a strong piece of work with significant potential impact, and addressing these points would further strengthen the manuscript’s clarity and scientific rigor.

The authors have satisfactorily addressed all of my questions and concerns and have revised the manuscript appropriately. Their exploration of MicroED/3D ED as an alternative approach for the direct analysis of protein crystals grown in cells constitutes a significant and intriguing contribution to the field.